# Computing Equilibrium beyond Unilateral Deviation

**Mingyang Liu, Gabriele Farina & Asuman Ozdaglar**
LIDS, EECS
Massachusetts Institute of Technology
Cambridge, MA 02139, USA
{liumy19,gfarina,asuman}@mit.edu

## ABSTRACT

Most familiar equilibrium concepts, such as Nash and correlated equilibrium, guarantee only that no single player can improve their utility by deviating unilaterally. They offer no guarantees against profitable coordinated deviations by coalitions. Although the literature proposes notions to address multilateral deviations (*e.g.*, strong Nash and coalition-proof equilibrium), these generally fail to exist. In this paper, we study a solution concept that accommodates multi-player deviations and is guaranteed to exist. We prove a fixed-parameter lower bound on the complexity of computing such an equilibrium and present an algorithm that matches this bound.

## 1 INTRODUCTION

Most equilibrium concepts studied so far, such as Nash equilibrium (NE) (Nash, 1950), correlated equilibrium (CE) (Aumann, 1974), coarse correlated equilibrium (CCE) (Moulin & Vial, 1978), and Stackelberg equilibrium (Von Stackelberg, 2010), guarantee only that *no individual player* can gain by deviating unilaterally. However, they offer no guarantees when multiple players deviate simultaneously by forming a coalition. In this paper, we address the following question:

> *What is an appropriate notion for capturing multilateral deviations, and how can it be computed?*

Previous notions that address coalition deviations, such as strong NE (Aumann, 1959)[1] and coalition-proof equilibrium (Bernheim et al., 1987), mostly fail to exist in general games (unlike NE). Therefore, instead of searching for a joint strategy immune to all coalition deviations, we focus on computing a joint strategy that *minimizes* the maximum average gain achievable by any coalition, which is the average of improvements over all its members. In other words, we can compute the most stable strategy profile, even if a perfectly stable one does not exist. We refer to this notion as the *Minimum Average-Strong Equilibrium* (MASE).

The difficulty of this optimization problem naturally depends on the complexity of the interactions between players. To formalize this, we introduce the Utility Dependency Graph, $\mathcal{G}(\mathcal{V}, \mathcal{E})$, where each player is a vertex. An edge connects two players, $i$ and $j$, if and only if there is some player $k$ whose utility is affected by the actions of both $i$ and $j$. Intuitively, an edge $(i, j)$ signifies that the actions of $i$ and $j$ are linked, as they jointly influence the payoff of some player $k$. This graph provides a clear map of the game's interaction structure, and its properties can help us understand the computational complexity of finding the MASE. For games with simple interaction structures (*e.g.*, a sparse Utility Dependency Graph), one might expect to compute the MASE efficiently.

However, computing the MASE is computationally challenging in the general case. We establish two key hardness results that delineate the problem's complexity.

First, the problem is fundamentally harder than finding equilibria like NE or CE. In those cases, we are solving a feasibility problem: finding a strategy where the maximum gain from deviating is at

---

[1] In this paper, we use the term *strong equilibrium* to broadly refer to any equilibrium concept that considers multilateral deviations.

most zero. For MASE, we must solve an optimization problem: minimizing this maximum gain. This distinction is crucial, and we show that even for the simplest case of single-player deviations (*i.e.*, coalitions of size one), approximating the MASE value to within a factor that is inverse polynomial in the number of players is NP-hard. This indicates that even without considering complex coalitions, the problem is intractable without additional assumptions on the game's structure.

Second, we show that this complexity is intrinsically tied to the structure of the Utility Dependency Graph. Building on the strong exponential time hypothesis (SETH) (Impagliazzo & Paturi, 2001) (see Theorem 4.3 for details), we prove that solving MASE requires time that is at least exponential in the treewidth[2] of the Utility Dependency Graph. This holds even when we only consider coalitions of a constant size. This result demonstrates that the treewidth is a fundamental barrier, and an exponential dependence on it is unavoidable.

Finally, we present a positive result that matches the lower-bound time complexity, up to exponential factors. We develop an algorithm that computes the MASE with a time complexity that is exponential in the treewidth of the Utility Dependency Graph. This demonstrates that our hardness result is tight and establishes the treewidth as the definitive parameter characterizing the complexity of computing the MASE. While the problem is hard in general, it becomes tractable for games where the underlying interaction structure is not too complex.

To summarize, our contributions are as follows:

1. Complexity characterization: We establish lower bounds on the computational complexity of computing the minimally deviated equilibrium, showing that the problem is inherently tied to the treewidth of the Utility Dependency Graph.
2. Algorithmic contribution: We design an algorithm that efficiently computes the minimally deviated equilibrium, achieving a running time that matches the established lower bound up to exponential dependence on treewidth.

## 2 PRELIMINARIES

For any vector $\boldsymbol{x} \in \mathbb{R}^n$, we use $x_i$ to denote its $i^{th}$ element and $\|\boldsymbol{x}\|_p$ to denote its $p$-norm. By default, $\|\boldsymbol{x}\|$ refers to the 2-norm. For a positive integer $N$, let $[N] := 1, 2, \ldots, N$. We denote the $(n-1)$-dimensional probability simplex by $\Delta^n := \{\boldsymbol{x} \in [0,1]^n : \sum_{i=1}^n x_i = 1\}$. More generally, for any discrete set $S$, let $|S|$ denote its cardinality and write $\Delta^S$ for the probability simplex over $S$, whose coordinates are indexed by elements of $S$ (*e.g.*, $\Delta^n = \Delta^{[n]}$). Similarly, $\mathbb{R}^S$ denotes the $|S|$-dimensional real vector space with coordinates indexed by $S$. For any set $S_1, S_2$, $S_1 \times S_2$ denotes the Cartesian product of sets $S_1$ and $S_2$. Finally, we let $\mathbb{1}(\text{argument})$ denote the indicator function, which equals 1 if the argument is true and 0 otherwise.

### 2.1 GAMES

A game is represented as a tuple $(N, \{\mathcal{A}_i\}_{i=1}^N, \{\mathcal{U}_i\}_{i=1}^N, \mathcal{S})$, where

- $N$ is the number of players.
- $\mathcal{A}_i$ is the action set of player $i$. For convenience, let $\mathcal{A} := \times_{i=1}^N \mathcal{A}_i$ denote the joint action set.
- $\mathcal{U}_i : \mathcal{A} \to [0,1]$ is the utility function of player $i \in [N]$.
- $\mathcal{S}$ is the set of coalitions, which is a set of subsets of players. For example, if only unilateral deviations are allowed (as in Nash equilibrium or coarse correlated equilibrium), then $\mathcal{S} = \{\{1\}, \{2\}, \ldots, \{N\}\}$.

For notational simplicity, for any subset of players $S \subseteq [N]$, we write $\mathcal{A}_S := \times_{i \in S} \mathcal{A}_i$. Throughout the paper, let $A := \max_{i \in [N]} |\mathcal{A}_i|$ denote the size of the largest action set.

For any joint action $\boldsymbol{a} \in \mathcal{A}$, let $a_i$ denote the action of player $i$, and let $\boldsymbol{a}_{-i} = (a_1, a_2, \ldots, a_{i-1}, a_{i+1}, \ldots a_N)$ be the joint action of all players except $i$. More generally, for any subset $S \subseteq [N]$, we write $\boldsymbol{a}_{-S}$ for the joint action of players outside $S$.

---

[2]Treewidth can be thought of as a formal measure of how sparse and "tree-like" a graph is.

|  | Confess (C) | Defect (D) |
|---|---|---|
| **Confess (C)** | $(0.6, 0.6)$ | $(0, 1)$ |
| **Defect (D)** | $(1, 0)$ | $(0.2, 0.2)$ |

Table 1: Utility matrix of the Prisoner's Dilemma. Each entry $(a, b)$ denotes the payoff of the row player $(a)$ and the column player $(b)$.

## 2.2 SUCCINCT REPRESENTATION

This paper focuses on multi-player games with a succinct representation. Specifically, each utility function $\mathcal{U}_i$ can be encoded using a number of bits polynomial in the number of players $N$, rather than requiring $\mathcal{O}\left(N \prod_{i=1}^{N} |\mathcal{A}_i|\right)$ bits as in the general case. Examples of succinctly represented games include polymatrix games (Howson Jr, 1972; Eaves, 1973) and congestion games (Rosenthal, 1973). Throughout the paper, we call an algorithm *efficient* if its running time is polynomial in $N$, as opposed to polynomial in $\prod_{i=1}^{N} |\mathcal{A}_i|$. We focus on succinct games because MASE can otherwise be solved by a linear program whose size grows exponentially with $N$ (see Section B). Moreover, the study of strong equilibrium is particularly compelling in large games, where exponential dependence on $N$ is computationally prohibitive.

## 3 MINIMUM AVERAGE-STRONG EQUILIBRIUM (MASE)

Several notions of strong equilibrium have been proposed, including the strong Nash equilibrium (NE) (Aumann, 1959), the sum-strong NE (Hoefer, 2013) (no improvement on the total gain of any coalition), and coalition-proof equilibrium (Bernheim et al., 1987). However, none of these exist in general games. To build intuition, we first illustrate why a strong NE does not exist in the Prisoner's Dilemma. We further show that the problem persists even when correlated strategies are allowed.

**Lemma 3.1.** In the Prisoner's Dilemma, no strong Nash nor strong correlated equilibrium exists when $\mathcal{S} = \{\{1\}, \{2\}, \{1, 2\}\}$ is the set of all non-empty subsets of players.

Since correlated equilibria include all Nash equilibria, it suffices to examine strong correlated equilibria. A strong correlated equilibrium is a correlated joint strategy where no subset of players (a coalition) can jointly deviate in a way that strictly improves the utility of all its members.

As shown in Table 1, any strategy with positive weight on $(C, C), (C, D), (D, C)$ yields a profitable deviation for at least one singleton coalition, $\{1\}$ or $\{2\}$. Conversely, placing all weight on $(D, D)$ creates a deviation to $(C, C)$ that benefits the coalition $\{1, 2\}$. Thus, no strong NE exists.

The failure of strong equilibria arises because coalition objectives may conflict, making it impossible to find a strategy that simultaneously satisfies all coalitions. Motivated by this, rather than requiring exact immunity to deviations, we instead seek to minimize the incentive to deviate since a minimizer always exists by Weierstrass theorem. This leads to the following definition of *Minimum Average-Strong Equilibrium* (MASE):

$$\pi^* \in \operatorname*{argmin}_{\pi \in \Delta^{\mathcal{A}}} \max_{S \in \mathcal{S}} \max_{\widehat{\boldsymbol{a}}_S \in \mathcal{A}_S} \frac{1}{|S|} \sum_{i \in S} \mathbb{E}_{\boldsymbol{a} \sim \pi} \left[ \mathcal{U}_i \left( \widehat{\boldsymbol{a}}_S, \boldsymbol{a}_{-S} \right) - \mathcal{U}_i \left( \boldsymbol{a} \right) \right]. \tag{MASE}$$

Intuitively, (MASE) selects the correlated strategy $\pi \in \Delta^{\mathcal{A}}$ that minimizes the maximum average gain attainable by any coalition across all possible coalitions. If this value is less than or equal to zero, then no coalition can simultaneously deviate in a way that yields a strictly positive total gain. Note that the algorithm presented in this paper can also be extended to handle any weighted average over the coalition.

A correlated strategy $\pi \in \Delta^{\mathcal{A}}$ is called an $\epsilon$-*MASE* if

$$\max_{S \in \mathcal{S}} \max_{\widehat{\boldsymbol{a}}_S \in \mathcal{A}_S} \frac{1}{|S|} \sum_{i \in S} \mathbb{E}_{\boldsymbol{a} \sim \pi} \left[ \mathcal{U}_i \left( \widehat{\boldsymbol{a}}_S, \boldsymbol{a}_{-S} \right) - \mathcal{U}_i \left( \boldsymbol{a} \right) \right]$$

$$\leq \max_{S \in \mathcal{S}} \max_{\widehat{\boldsymbol{a}}_S \in \mathcal{A}_S} \frac{1}{|S|} \sum_{i \in S} \mathbb{E}_{\boldsymbol{a} \sim \pi^*} \left[ \mathcal{U}_i \left( \widehat{\boldsymbol{a}}_S, \boldsymbol{a}_{-S} \right) - \mathcal{U}_i \left( \boldsymbol{a} \right) \right] + \epsilon.$$

A Strong Nash equilibrium requires that for any deviating coalition, at least one member does not strictly improve their utility. In contrast, $\epsilon$-MASE aims to minimize the average improvement over all players within any given coalition. From another perspective, $\epsilon$-MASE minimizes the incentive to deviate, even when coalition members can freely reallocate utility within the coalition.

## 4    HARDNESS OF SOLVING MASE

Recall that we call an algorithm *efficient* if it runs in time polynomial in $N$. In this section, we first establish the computational hardness of computing $\epsilon$-MASE.

**Theorem 4.1.** Computing $\epsilon$-MASE is NP-hard, even when $\mathcal{S}$ only contains singletons (coalitions of size one) and $1/\epsilon$ is polynomial in the number of players.

The proof is deferred to Section C. Importantly, Theorem 4.1 highlights a fundamental distinction from the case of CCE, which can be computed efficiently (Papadimitriou & Roughgarden, 2008). The reason is that for CCE it suffices to find a correlated strategy $\pi \in \Delta^{\mathcal{A}}$ such that the deviation gap, $\max_{i \in [N]} \max_{\widehat{a}_i \in \mathcal{A}_i} \mathbb{E}_{\boldsymbol{a} \sim \pi} [\mathcal{U}_i(\widehat{a}_i, \boldsymbol{a}_{-i}) - \mathcal{U}_i(\boldsymbol{a})]$, is less or equal to zero, whereas here we must find a strategy that minimizes the gap. Together with the linear programming characterization in Section B, this implies that computing $\epsilon$-MASE is actually NP-complete. In fact, Anagnostides et al. (2025) recently showed that even minimizing the *average* deviation gap of CCE across all players (instead of the maximum gap considered here) is also NP-complete.

### 4.1    FIXED PARAMETER LOWER BOUND

Next, we present a more refined hardness result: a fixed-parameter lower bound for computing MASE. To do so, we first formalize the notion of dependencies among players' utilities.

For each player $i \in [N]$, define the relevant set $\mathcal{N}(i) \subseteq [N]$ consisting of all players $j \in [N]$ (including $j = i$) such that the action of $j$ can affect the utility of $i$. Formally, $j \in [N]$ is in $\mathcal{N}(i)$ if and only if there exist $\boldsymbol{a}_{-j} \in \mathcal{A}_{-j}$ and $a_j, a_j' \in \mathcal{A}_j$ such that $\mathcal{U}_i(a_j, \boldsymbol{a}_{-j}) \neq \mathcal{U}_i(a_j', \boldsymbol{a}_{-j})$. This leads to the following graph representation.

**Definition 4.2** (Utility Dependency Graph). The utility dependence graph $\mathcal{G} = (\mathcal{V}, \mathcal{E})$ is an undirected graph with vertex set $\mathcal{V} = [N]$ representing the players, and edge set $\mathcal{E} = \bigcup_{k \in [N]} \{(i, j) \mid i, j \in \mathcal{N}(k), i \neq j\}$.

Since $\mathcal{U}_i$ depends only on the actions of players in $\mathcal{N}(i)$, we may equivalently write $\mathcal{U}_i(\boldsymbol{a}_C) = \mathcal{U}_i(\boldsymbol{a}_C, \boldsymbol{a}_{-C}')$ for arbitrary $\boldsymbol{a}_{-C}' \in \mathcal{A}_{-C}$, where $C \supseteq \mathcal{N}(i)$. It is worth noting that this definition differs from the graph of a graphical game (Kakade et al., 2003; Kearns et al., 2001). Here, players $i$ and $j$ are connected if both influence the utility of some other player $k$, even if $i$ and $j$ do not directly affect each other. Whereas in graphical games, two players $i$ and $j$ are connected if and only if at least one can influence the other's utility.

With this graph structure in place, we can connect the hardness of computing MASE to the treewidth of $\mathcal{G}$. Intuitively, treewidth measures how close a graph is to being a tree: the treewidth of $\mathcal{G}$ is one when $\mathcal{G}$ is a tree, and it is $N-1$ when $\mathcal{G}$ is a complete graph. Throughout this section, let $\mathcal{O}^*$ denote asymptotic complexity with factors polynomial in $N$ suppressed.

**Theorem 4.3** (Treewidth). Suppose a tree decomposition of the Utility Dependency Graph is given. Under the Strong Exponential Time Hypothesis (SETH) (Impagliazzo & Paturi, 2001),[3] (MASE) cannot be computed in $\mathcal{O}^*((A - \zeta)^{\text{tw}(\mathcal{G})})$ for any $\zeta > 0$. Moreover, under the additional assumption that BPP=P,[4] $\frac{1}{9N^2}$-approximate MASE cannot be computed in $\mathcal{O}^*((A - \zeta)^{\text{tw}(\mathcal{G})})$ for any $\zeta > 0$.

The proof is deferred to Section C.2. For approximate MASEs we assume BPP=P, which is standard in the literature (Arora & Barak, 2009), because the reduction involves sampling joint actions from

---

[3]SETH assumes that SAT cannot be solved in $\mathcal{O}^*((2 - \zeta)^n)$ for any $\zeta > 0$, where $n$ is the number of variables in the SAT instance.

[4]This assumption implies that any problem with a polynomial-time randomized algorithm also has a polynomial-time deterministic algorithm.

the approximate MASE. Since enumerating all $\boldsymbol{a} \in \mathcal{A}$ is computationally infeasible, we rely on randomized sampling. This yields only a randomized algorithm for the original `NP-hard` problem, and the assumption `BPP=P` ensures that such a randomized algorithm can be derandomized into a deterministic one, completing the reduction.

Theorem 4.3 shows that the computational complexity of solving MASE is inherently tied to the treewidth of the Utility Dependency Graph. Intuitively, when the treewidth is large, each player's utility depends on many others, making even the evaluation of coalition deviations computationally demanding (enumerating over all $\widehat{\boldsymbol{a}}_S \in \mathcal{A}_S$). In contrast, when the treewidth is small, such as zero (each player's utility depends only on their own action), computing MASE becomes trivial, since each player's utility can be maximized independently. In polymatrix games (Eaves, 1973), the treewidth of the Utility Dependency Graph can be bounded by that of its corresponding graph. Further details are provided in Section I.

# 5    EFFICIENT COMPUTATION OF MASE

Although an (MASE) lives in an exponentially large space (of size $|\mathcal{A}|$), it can still be computed efficiently. This is because the equilibrium always admits a compact representation.

**Theorem 5.1** (Efficient Representation). For any $\epsilon \geq 0$, at least one of the $\epsilon$-MASE can be represented as a linear combination of $\sum_{S \in \mathcal{S}} |S| \cdot A^{\mathrm{tw}(\mathcal{G})+1}$ pure strategies, where $\mathrm{tw}(\mathcal{G})$ is the treewidth of Utility Dependency Graph.

The proof is deferred to Section D. Intuitively, Theorem 5.1 shows that there must be an $\epsilon$-MASE that always has a sparse representation. Since a pure strategy can be encoded by the index of its unique action with nonzero probability, this compactness makes computation tractable.

## 5.1    META-GAME BETWEEN THE CORRELATOR AND DEVIATOR

To compute an (MASE), we reformulate the problem as a *meta-game* between two players: the *correlator* and the *deviator* (Hart & Schmeidler, 1989). The correlator chooses the correlated strategy $\pi \in \Delta^{\mathcal{A}}$, while the deviator selects deviations. The game is zero-sum: the correlator aims to minimize the coalition's gain from deviation, and the deviator aims to maximize it. Formally:

$$\min_{\pi \in \Delta^{\mathcal{A}}} \max_{\mu \in \Delta^{\times_{S \in \mathcal{S}} \mathcal{A}_S}} F(\pi, \mu), \tag{5.1}$$

where

$$F(\pi, \mu) := \sum_{S \in \mathcal{S}} \sum_{\widehat{\boldsymbol{a}}_S \in \mathcal{A}_S} \frac{\mu(S, \widehat{\boldsymbol{a}}_S)}{|S|} \sum_{i \in S} \mathbb{E}_{\boldsymbol{a} \sim \pi} \left[ \mathcal{U}_i \left( \widehat{\boldsymbol{a}}_S, \boldsymbol{a}_{-S} \right) - \mathcal{U}_i \left( \boldsymbol{a} \right) \right]. \tag{5.2}$$

Here, we extend the deviator's decision space from a discrete to a continuous set. This relaxation does not strengthen the deviator, since the objective is linear in $\mu$, and the maximum is always attained at an extreme point. Therefore, (5.1) is equivalent to the original definition in (MASE).

A natural idea is to apply no-regret learning algorithms simultaneously for the correlator and deviator. However, directly updating the full distributions $\pi$ and $\mu$ is infeasible, because the underlying spaces are exponentially large.

Fortunately, Theorem 5.1 implies that maintaining the full distributions is unnecessary: it suffices to keep track of a polynomial number of pure strategies, and use their convex combination as the approximate equilibrium. This motivates our use of *Follow the Perturbed Leader* (FTPL) (Hazan et al., 2016), where each decision at a timestep is a pure strategy, which can be represented compactly.

Let $\pi^{(t)} \in \Delta^{\mathcal{A}}$ and $\mu^{(t)} \in \Delta^{\times_{S \in \mathcal{S}} \mathcal{A}_S}$ denote the decision variables at timestep $t \geq 1$ for the correlator and the deviator, respectively. The interaction between these two players can be described

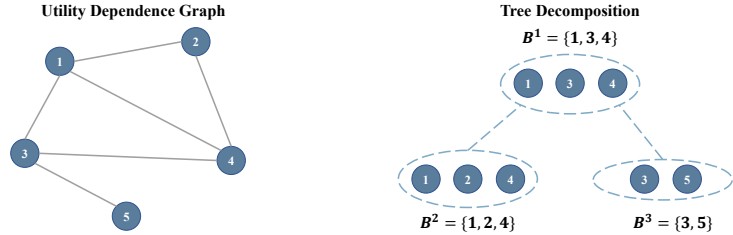

Figure 1: An illustration of a tree decomposition of the Utility Dependency Graph.

by the update rule

$$
\begin{aligned}
\pi^{(t+1)} &\in \underset{\pi \in \Delta^{\mathcal{A}}}{\operatorname{argmin}} \sum_{\tau=1}^{t} F\left(\pi, \mu^{(\tau)}\right) - \left\langle \widetilde{\boldsymbol{n}}^{(t+1)}, \pi \right\rangle \\
\mu^{(t+1)} &\in \underset{\mu^{(t)} \in \Delta^{\times_{S \in \mathcal{S}} \mathcal{A}_S}}{\operatorname{argmax}} \sum_{\tau=1}^{t} F\left(\pi^{(\tau)}, \mu\right) + \left\langle \widetilde{\boldsymbol{m}}^{(t+1)}, \mu \right\rangle,
\end{aligned}
\tag{5.3}
$$

where $\widetilde{\boldsymbol{n}}^{(t+1)}$ and $\widetilde{\boldsymbol{m}}^{(t+1)}$ are noise vectors sampled independently at each timestep from some distribution, which we will specify later. These noise terms play the role of regularizers in on-line mirror descent (OMD) (Hazan et al., 2016), ensuring stability in the updates by controlling $\mathbb{E}\left[\left\|\pi^{(t+1)} - \pi^{(t)}\right\|\right]$ and $\mathbb{E}\left[\left\|\mu^{(t+1)} - \mu^{(t)}\right\|\right]$.

Since $F(\pi, \mu)$ is bilinear in $(\pi, \mu)$, both the minimization and maximization problems admit solutions at vertices of their respective decision spaces. In other words, the argmin for the correlator and the argmax for the deviator always contain at least one pure strategy.

In what follows, we will explain in detail how to update $\pi$ efficiently under this framework. The update of $\mu$ is deferred to Section G.

## 5.2 EFFICIENT UPDATE OF $\pi$

The key step in updating $\pi^{(t+1)}$ is to select a pure strategy, *i.e.*, a joint action $\boldsymbol{a}^{(t+1)} \in \mathcal{A}$ with $\pi^{(t+1)}(\boldsymbol{a}^{(t+1)}) = 1$, that minimizes the objective. To gain insight into this update rule, we first examine how to compute $\operatorname{argmin}_{\pi \in \Delta^{\mathcal{A}}} F(\pi, \mu)$ for a fixed $\mu$.

Suppose we want to find a joint action $\widetilde{\boldsymbol{a}} \in \mathcal{A}$ such that the pure strategy $\widetilde{\pi}$ with $\widetilde{\pi}(\widetilde{\boldsymbol{a}}) = 1$ minimizes $F(\widetilde{\pi}, \mu)$. Expanding the definition, we obtain

$$
\begin{aligned}
F(\widetilde{\pi}, \mu) &= \sum_{i=1}^{N} \sum_{S \in \mathcal{S}: \, i \in S} \sum_{\widehat{\boldsymbol{a}}_S \in \mathcal{A}_S} \frac{\mu(S, \widehat{\boldsymbol{a}}_S)}{|S|} \left(\mathcal{U}_i\left(\widehat{\boldsymbol{a}}_S, \widetilde{\boldsymbol{a}}_{-S}\right) - \mathcal{U}_i\left(\widetilde{\boldsymbol{a}}\right)\right) \\
&= \sum_{i=1}^{N} \sum_{S \in \mathcal{S}: \, i \in S} \sum_{\widehat{\boldsymbol{a}}_S \in \mathcal{A}_S} \frac{\mu(S, \widehat{\boldsymbol{a}}_S)}{|S|} \left(\mathcal{U}_i\left(\widehat{\boldsymbol{a}}_{S \cap \mathcal{N}(i)}, \widetilde{\boldsymbol{a}}_{\mathcal{N}(i) \setminus S}\right) - \mathcal{U}_i\left(\widetilde{\boldsymbol{a}}_{\mathcal{N}(i)}\right)\right).
\end{aligned}
$$

Therefore, for each candidate $\widetilde{\boldsymbol{a}} \in \mathcal{A}$, only the local actions $\widetilde{\boldsymbol{a}}_{\mathcal{N}(i)}$ matter for the expression above. If we can evaluate this expression efficiently,[5] then for each player $i \in [N]$ we may search for $\widetilde{\boldsymbol{a}}_{\mathcal{N}(i)}$ that minimizes it. However, a difficulty arises because $\mathcal{N}(i)$ and $\mathcal{N}(j)$ may overlap across different players. Hence, we must ensure that the local assignments remain globally consistent.

To address this, we now introduce the concept of a tree decomposition and show how it enables us to optimize $F$ efficiently. Throughout the paper, we assume that a tree decomposition is given, and analyze the complexity only with respect to this decomposition.

**Tree decomposition.** A tree decomposition $\mathcal{T} := B^1, B^2, \ldots, B^K$ of the Utility Dependency Graph $\mathcal{G} = (\mathcal{V}, \mathcal{E})$ is a tree with $K$ nodes (bags), each $B^k \subseteq \mathcal{V}$ where $\mathcal{V} = [N]$, satisfying the following properties (Diestel, 2025):

---

[5]This is possible since $\mu$ is a linear combination of pure strategies when updated according to (5.3).

1. $\bigcup_{k=1}^{K} B^k = [N]$.
2. For every edge $(i, j) \in \mathcal{E}$, there exists $k$ with $\{i, j\} \subseteq B^k$.
3. For any player $i \in [N]$, if $i$ appears in two bags $B, B' \in \mathcal{T}$, then every bag on the path from $B$ to $B'$ also contains $i$.

As illustrated in Figure 1, the tree decomposition separates the game into overlapping bags. For example, since $B^2$ and $B^3$ only overlap at $B^1$, then $B^2$ and $B^3$ can be optimized independently, with consistency later enforced at $B^1$.

Since any clique in $\mathcal{G}$ is contained in some bag (Diestel, 2025), for every player $i \in [N]$ there exists a bag $B$ with $\mathcal{N}(i) \subseteq B$. We arbitrarily assign each player $i$ to such a bag.

**Dynamic programming on the tree.** We begin by choosing an arbitrary bag as the root of the tree decomposition and denote it by $B^r$. For each bag $B \in \mathcal{T}$, let $C(B)$ denote the set of its children. With this setup, we maintain a vector $\boldsymbol{d}^{(t+1)} \in \mathbb{R}^{\times_{B \in \mathcal{T}} \mathcal{A}_B}$, defined as

$$
d^{(t+1)}(B, \boldsymbol{a}_B) = \sum_{\tau=1}^{t} \sum_{S \in \mathcal{S}} \frac{1}{|S|} \sum_{\substack{i \in S: \\ i \text{ assigned to } B}} \sum_{\widehat{\boldsymbol{a}}_S \in \mathcal{A}_S} \mu^{(\tau)}(S, \widehat{\boldsymbol{a}}_S) \left( \mathcal{U}_i \left( (\widehat{\boldsymbol{a}}_{S \cap B}, \boldsymbol{a}_{B \setminus S}) \right) - \mathcal{U}_i(\boldsymbol{a}_B) \right)
$$
$$
+ \sum_{B' \in C(B)} \min_{\substack{\boldsymbol{a}'_{B'} \in \mathcal{A}_{B'}: \\ \boldsymbol{a}_{B \cap B'} = \boldsymbol{a}'_{B \cap B'}}} d^{(t+1)}(B', \boldsymbol{a}'_{B'}) - n^{(t+1)}(B, \boldsymbol{a}_B),
$$
(5.4)

where $n^{(t+1)}(B, \boldsymbol{a}_B) \sim \text{Exp}(\eta)^6$ is sampled from an exponential distribution. Therefore, in (5.3), $\widetilde{n}^{(t+1)}(\boldsymbol{a}) = \sum_{B \in \mathcal{T}} n^{(t+1)}(B, \boldsymbol{a}_B)$. Since each $i$ assigned to $B$ satisfies $\mathcal{N}(i) \subseteq B$, the utility $\mathcal{U}_i(\boldsymbol{a}_B)$ can be written in terms of $\boldsymbol{a}_B$ alone. Moreover, the summation $\sum_{\widehat{\boldsymbol{a}}_S \in \mathcal{A}_S}$ can be computed efficiently, since $\mu^{(\tau)}$ is updated via (5.3) and is therefore a pure strategy.

**Reconstructing the strategy.** The optimal joint action $\boldsymbol{a}^{(t+1)} \in \mathcal{A}$ is then reconstructed recursively from the root $B^r$ to the leaves:

$$
\boldsymbol{a}_{B^r}^{(t+1)} = \underset{\boldsymbol{a}_{B^r} \in \mathcal{A}_{B^r}}{\arg\min} \; d^{(t+1)}(B^r, \boldsymbol{a}_{B^r})
$$
$$
\forall B \in C(B^r), \quad \boldsymbol{a}_{B \setminus B^r}^{(t+1)} = \underset{\boldsymbol{a}_{B \setminus B^r} \in \mathcal{A}_{B \setminus B^r}}{\arg\min} \; d^{(t+1)} \left( B, (\boldsymbol{a}_{B \setminus B^r}, \boldsymbol{a}_{B \cap B^r}^{(t+1)}) \right).
$$
(5.5)

By **Property 1 of Tree Decomposition**, every player's action will be included. Since $\arg\min_{\boldsymbol{a}_{B \setminus B^r} \in \mathcal{A}_{B \setminus B^r}}$ is taken over $\mathcal{A}_{B \setminus B^r}$, no contradictions arise by **Property 3 of Tree Decomposition**. We then set $\pi^{(t+1)}(\boldsymbol{a}^{(t+1)}) = 1$.

The regret bound of this procedure is summarized below.

**Theorem 5.2.** Consider (5.3). For any $\delta > 0$, with probability at least $1 - \delta$, the following holds:

$$
\max_{\widehat{\pi} \in \Delta^{\mathcal{A}}} \sum_{t=1}^{T} F\left(\pi^{(t)}, \mu^{(t)}\right) - F\left(\widehat{\pi}, \mu^{(t)}\right) \leq 2|\mathcal{T}| \frac{1 + (\text{tw}(\mathcal{G}) + 1) \log A}{\eta} + 2\eta|\mathcal{T}|T + \sqrt{2T \log \frac{1}{\delta}}.
$$

The proof is given in Section F. Importantly, Theorem 5.2 shows that by setting $\eta = 1/\sqrt{T}$, we obtain $\mathcal{O}(\sqrt{T})$ regret. Since the update rule for $\mu$ mirrors that of $\pi$, the detailed analysis is deferred to Section G. We now formally state the regret bound for $\mu$ in the following theorem.

**Theorem 5.3.** Consider the updates in (5.3). For any $\delta > 0$, with probability at least $1 - \delta$, the following holds:

$$
\max_{\widehat{\mu} \in \Delta^{\times_{S \in \mathcal{S}} \mathcal{A}_S}} \sum_{t=1}^{T} F\left(\pi^{(t)}, \widehat{\mu}\right) - F\left(\pi^{(t)}, \mu^{(t)}\right) \leq 2|\mathcal{T}| \frac{1 + (\text{tw}(\mathcal{G}) + 1) \log A}{\eta} + 2\eta|\mathcal{T}|T + \sqrt{2T \log \frac{1}{\delta}}.
$$

The complete proof is provided in Section G.

---

$^6 \Pr(x \geq w) = \exp(-\eta w)$ when $x \sim \text{Exp}(\eta)$.

## 5.3 Computation of Equilibrium

For any $\delta' > 0$, by setting $\delta = \frac{\delta'}{2}$ in Theorem 5.2 and Theorem 5.3, and applying the union bound, we obtain that with probability at least $1 - \delta'$, the following holds:

$$\max_{\widehat{\mu} \in \Delta^{\times_{S \in \mathcal{S}} \mathcal{A}_S}} \sum_{t=1}^{T} F\left(\pi^{(t)}, \mu\right) - \min_{\widehat{\pi} \in \Delta^{\mathcal{A}}} \sum_{t=1}^{T} F\left(\widehat{\pi}, \mu^{(t)}\right)$$

$$\leq 4\left|\mathcal{T}\right| \frac{1 + (\mathrm{tw}(\mathcal{G}) + 1) \log A}{\eta} + 4\eta\left|\mathcal{T}\right| T + 2\sqrt{2T \log \frac{2}{\delta'}}. \tag{5.6}$$

We now connect this bound to the convergence of the average strategy profile. Let $\pi^*, \mu^*$ be the solution to (5.1), and define the average strategies $\bar{\pi} := \frac{1}{T} \sum_{t=1}^{T} \pi^{(t)}$ and $\bar{\mu} := \frac{1}{T} \sum_{t=1}^{T} \mu^{(t)}$. The left-hand side of (5.6) corresponds to the duality gap: $\max_{\widehat{\mu} \in \Delta^{\times_{S \in \mathcal{S}} \mathcal{A}_S}} F(\bar{\pi}, \widehat{\mu}) - \min_{\widehat{\pi} \in \Delta^{\mathcal{A}}} F(\widehat{\pi}, \bar{\mu})$. Since $\pi^*, \mu^*$ are optimal solutions to (5.6), they satisfy

$$\min_{\widehat{\pi} \in \Delta^{\mathcal{A}}} F(\widehat{\pi}, \bar{\mu}) \leq F(\pi^*, \mu^*) \leq \max_{\widehat{\mu} \in \Delta^{\times_{S \in \mathcal{S}} \mathcal{A}_S}} F(\bar{\pi}, \widehat{\mu}).$$

Combining these pieces, we arrive at the following finite-time convergence guarantee:

**Theorem 5.4.** Let $\pi^*, \mu^*$ be the solution of (5.1), and define $\bar{\pi} := \frac{1}{T} \sum_{t=1}^{T} \pi^{(t)}$, $\bar{\mu} := \frac{1}{T} \sum_{t=1}^{T} \mu^{(t)}$. Then, for any $\delta > 0$, with probability at least $1 - \delta$, we have

$$\max_{\widehat{\mu} \in \Delta^{\times_{S \in \mathcal{S}} \mathcal{A}_S}} F(\bar{\pi}, \widehat{\mu}) \leq F(\pi^*, \mu^*) + 4\left|\mathcal{T}\right| \frac{1 + (\mathrm{tw}(\mathcal{G}) + 1) \log A}{\eta T} + 4\eta\left|\mathcal{T}\right| + 2\sqrt{\frac{2 \log \frac{2}{\delta}}{T}}.$$

With $\eta = \frac{1}{\sqrt{T}}$, the average strategy $\bar{\pi}$ constitutes an $\mathcal{O}\left(\frac{\left|\mathcal{T}\right| \cdot \mathrm{tw}(\mathcal{G}) \log A + \sqrt{\log \frac{2}{\delta}}}{\sqrt{T}}\right)$-MASE. The overall running time is $\mathcal{O}\left(T \cdot \left|\mathcal{S}\right| \cdot \left|\mathcal{T}\right| \cdot A^{\mathrm{tw}(\mathcal{G})+1}\right)$. Hence, the exponential dependence aligns with the lower bound in Theorem 4.3.

## 6 Experiments

In this section, we compare our algorithm against several baselines: Follow the Regularized Leader with a Euclidean regularizer (FTRL), Hedge, Follow the Perturbed Leader with an exponential noise distribution (FTPL; all players run FTPL independently), and Online Mirror Descent with a Euclidean regularizer (OMD) (Hazan et al., 2016). We also plot the ground-truth MASE computed via linear programming (LP) in Section B.

We evaluate the algorithms on three criteria:

- **Exploitability.** $(\max_{i \in [N]} \max_{\widehat{a}_i \in \mathcal{A}_i} \mathbb{E}_{\boldsymbol{a} \sim \pi}[\mathcal{U}_i(\widehat{a}_i, \boldsymbol{a}_{-i}) - \mathcal{U}_i(\boldsymbol{a})])$: the maximum gain a single player can obtain by deviating unilaterally. Exploitability $\leq 0$ indicate a Nash equilibrium (or a correlated equilibrium if $\pi$ is correlated).
- **Coalition exploitability.** $(\max_{\mu \in \Delta^{\times_{S \in \mathcal{S}} \mathcal{A}_S}} F(\pi, \mu))$: the maximum average gain when a coalition deviates simultaneously. We take $\mathcal{S}$ to be the set of all non-empty player subsets.
- **Social welfare.** $(\sum_{i=1}^{N} \mathbb{E}_{\boldsymbol{a} \sim \pi}[\mathcal{U}_i(\boldsymbol{a})])$: the sum of all players' utilities.

Utility definitions and additional details are provided in Section H. In the Prisoner's Dilemma (Luce & Raiffa, 1957), the MASE corresponds to players choosing $(C, D)$ and $(D, C)$ with probability $0.5$ each, yielding a social welfare of $1.0$. In contrast, because the unique NE/CCE in this game is $(D, D)$, the baselines converge to that outcome, with a lower social welfare of $0.4$. Thus, in the Prisoner's Dilemma, MASE promotes cooperation and achieves higher utility.

In the Stag Hunt, there are two Nash equilibria, one of which attains higher utility. As shown in Figure 2, all baselines converge to the worse equilibrium, whereas MASE converges to the better one. Finally, in terms of exploitability (unilateral deviations), MASE remains close to the baselines, while the baselines are substantially more fragile to multilateral deviations.

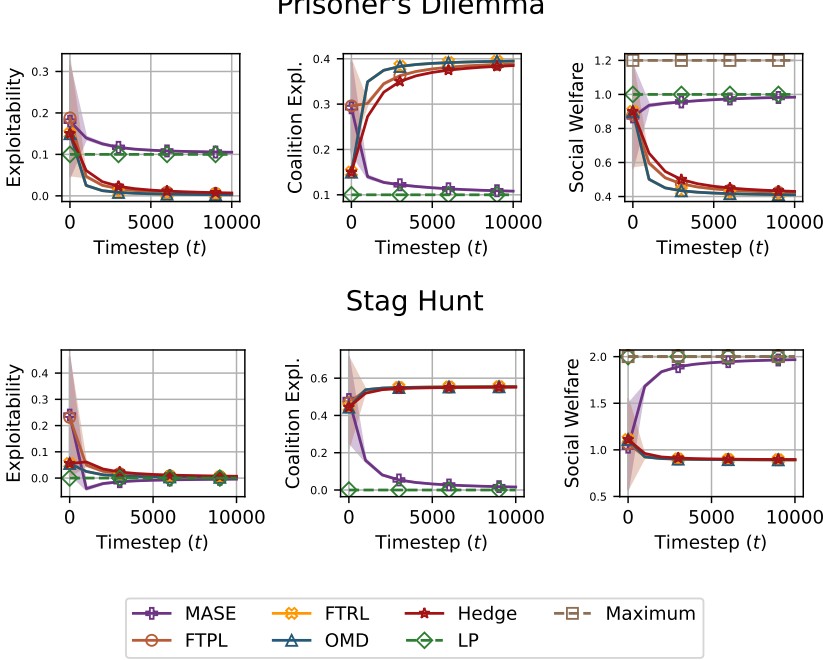

Figure 2: LP denotes the linear programming solution from Section B, and Maximum denotes the maximum achievable social welfare. The baselines are comparatively fragile to multilateral deviations, while MASE is more robust and achieves higher social welfare. At the same time, MASE's exploitability is close to that of the baselines.

## 6.1 TRADE-OFF BETWEEN EXPLOITABILITY AND SOCIAL WELFARE

In Figure 2, we can see that by allowing exploitability to increase from $0.0$ to $0.1$, the social welfare of MASE increases from $0.4$ to $1.0$. This raises a natural question:

> Given a tolerance $\epsilon \geq 0$, what is the maximum social welfare achievable by an equilibrium with exploitability at most $\epsilon$?

In other words, if we are willing to sacrifice equilibrium robustness, how much can we improve social welfare? Interestingly, this trade-off can be computed efficiently using a variant of our MASE framework. Specifically, we solve the following weighted objective:

$$\operatorname*{argmin}_{\pi \in \Delta^{\mathcal{A}}} \max_{S \in \mathcal{S}} \max_{\widehat{\boldsymbol{a}}_S \in \mathcal{A}_S} \frac{w_S}{|S|} \sum_{i \in S} \mathbb{E}_{\boldsymbol{a} \sim \pi} \left[ \mathcal{U}_i \left( \widehat{\boldsymbol{a}}_S, \boldsymbol{a}_{-S} \right) - \mathcal{U}_i \left( \boldsymbol{a} \right) \right], \tag{6.1}$$

where $\boldsymbol{w} \in \mathbb{R}^{\mathcal{S}}$ is a vector of non-negative weights. We have the following lemma.

**Lemma 6.1.** For any $\epsilon > 0$, computing the CCE with exploitability no more than $\epsilon$ that maximizes social welfare is equivalent to (6.1) by setting $\mathcal{S} = \{\{i\}\}_{i \in [N]} \cup \{[N]\}$ and using the weights:

$$w_S = \begin{cases} w & \text{if } |S| = 1 \\ 1 - w & \text{if } S = [N] \end{cases}$$

for some $w \in [0, 1)$. Conversely, solving (6.1) with these parameters corresponds to finding a point on the Pareto frontier of social welfare and exploitability.

The proof is postponed to Section H.4. With Lemma 6.1, we can compute the Pareto frontier by solving (6.1) for different values of $w$. The results are shown in Figure 3. In the Stag Hunt, since one of the Nash equilibria already maximizes social welfare, the social welfare remains fixed at its optimal value for all $w \in [0, 1)$.

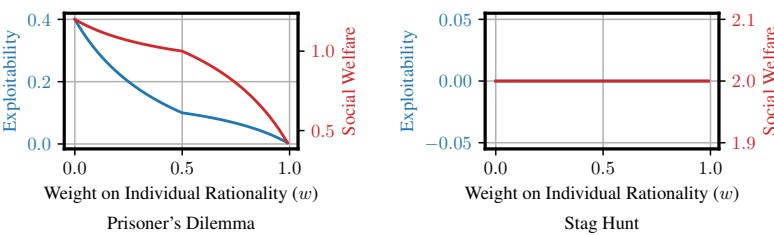

Figure 3: The trade-off between exploitability and social welfare in the Prisoner's Dilemma and the Stag Hunt.

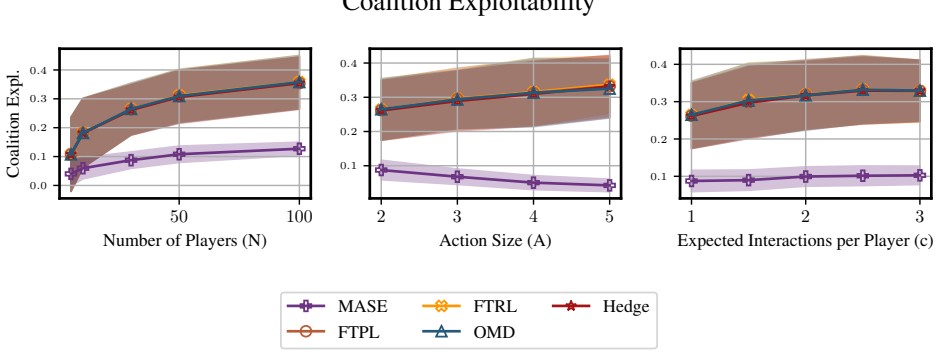

Figure 4: The coalition exploitability of random polymatrix games of different sizes when coalitions with no more than two players are considered. A larger expected number of interactions per player ($c$) generally corresponds to a larger treewidth of the Utility Dependency Graph.

## 6.2 COALITION EXPLOITABILITY IN LARGER GAMES

As shown in Figure 4, the coalition exploitability of the average strategy generated by classical no-regret learning algorithms increases as the game size grows. Note that we only consider coalitions of size no more than two. This trend underscores the importance of minimizing coalition exploitability. As games become larger, the equilibria to which these algorithms converge become increasingly fragile to coalition deviations, necessitating approaches that explicitly account for such multilateral deviations. Further details are provided in Section I.

## 7 CONCLUSION

In this work, we introduced the Minimum Average-Strong Equilibrium (MASE), a tractable solution concept that accounts for multilateral deviations by minimizing each coalition's average incentive to deviate. We established that computing an approximate MASE is NP-hard even with singleton coalitions and proved a fixed-parameter lower bound showing unavoidable exponential dependence on the treewidth of the Utility Dependency Graph. We then designed an algorithm—combining a correlator–deviator meta-game with FTPL updates and dynamic programming over a tree decomposition, whose running time matches this lower bound up to the treewidth factor. Empirically, MASE is substantially more robust to coalition deviations than standard baselines while improving social welfare in canonical games, all without materially worsening unilateral exploitability.

In the future, it is natural to move beyond uniform averaging within coalitions. A compelling open direction is to characterize lower and upper bounds for objectives that minimize the minimal incentive within each coalition. More broadly, extending these ideas to richer coalition objectives would mature the strong-equilibrium framework and yield more solution concepts that go beyond unilateral deviations.

## ACKNOWLEDGEMENT

The authors are grateful to the anonymous reviewers and the area chair for their constructive feedback. M.L. was supported by the MathWorks Fellowship. A.O. and G.F. were supported in part by the ONR grant N000142512296. G.F. was additionally supported by CCF-2443068 and an AI2050 Early Career Fellowship.

## 8 ETHICS STATEMENT

This paper presents work that aims to advance the field of game theory. There are many potential societal consequences of our work, none of which we feel must be specifically highlighted here.

## 9 REPRODUCIBILITY STATEMENT

The proof and assumptions are stated in Sections C, D, F and G.2.

## 10 USE OF LARGE LANGUAGE MODELS

In this paper, we use large language models (LLMs). For example, by correcting grammatical errors, identifying related work to reduce the risk of overlooking relevant papers, helping create illustrative figures (Figure 1), and assisting with coding.

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

## A   RELATED WORK

In this section, we review the literature on strong equilibrium from three perspectives: existence, time complexity, and computation.

**Existence of Strong Equilibrium.**   Aumann (1959) introduced the strong NE, where no coalition (a nonempty subset of players) can deviate in a way that strictly improves the utility of all its members. However, even in simple two-player games such as the Prisoner's Dilemma (Luce & Raiffa, 1957), a strong NE does not exist when players can coordinate. To address this, Bernheim et al. (1987) proposed the coalition-proof equilibrium, which restricts the set of deviations. Yet, this concept also fails to guarantee existence, already in three-player games (Bernheim et al., 1987). More recently, Rahn & Schäfer (2015) studied the notion of $\alpha$-approximate $k$-equilibrium, where no coalition of size at most $k$ can deviate to improve each member's utility by at least a factor of $\alpha \geq 1$. They further showed that such equilibria exist in graph coordination games only under specific conditions, for instance, when $\alpha \geq 2$. Motivated by these non-existence results, we instead focus on minimizing the maximum average gain from coalition deviations (MASE), a quantity that is always well defined.

**Complexity of Strong Equilibrium.**   Since a strong NE degenerates to an NE when only singleton coalitions are considered, computing a strong NE is `PPAD-hard` in general (Daskalakis et al., 2009; Chen & Deng, 2006). Beyond computation, Conitzer & Sandholm (2008) showed that even deciding whether a strong NE exists is `NP-complete` in two-player symmetric games, and Berthelsen & Hansen (2022) further established that the problem is `∃ℝ-complete` for three-player games. Similarly, Rahn & Schäfer (2015) proved that determining the existence of a strong NE is `NP-complete` in graph coordination games, even when restricting attention to coalitions of constant size. To the best of our knowledge, however, no hardness results are known for computing strong equilibria when correlation on the joint strategy is allowed, *i.e.*, a correlated strategy immune to coalition deviations. In this paper, we show that computing a correlated strategy that minimizes the average gain from coalition deviations is `NP-hard`. Moreover, we establish a fixed-parameter lower bound based on the treewidth of the Utility Dependency Graph, demonstrating an inherent computational barrier in solving the MASE considered here.

**Computation of Strong Equilibrium.**   Holzman & Law-Yone (1997) and Rozenfeld & Tennenholtz (2006) developed algorithms to compute strong NE and correlated strong equilibria in congestion games under certain conditions in polynomial time. Rahn & Schäfer (2015) showed that a strong NE can also be computed in polynomial time when the graph coordination game is defined on a tree. In contrast, Gatti et al. (2013) proposed a spatial branch-and-bound algorithm for computing strong NE more generally, but its runtime is exponential. Along the same lines, Nessah & Tian (2014) also provided a computationally intractable algorithm. Of independent interest, Papadimitriou & Roughgarden (2008) introduced an efficient algorithm for computing optimal CE—*e.g.*, a CE that maximizes the social welfare—in graphical games (Kearns et al., 2001; Kakade et al., 2003) with bounded treewidth, using linear programming. In this paper, we develop a new algorithm for computing MASE based on no-regret learning, with a time complexity that matches the lower bound dictated by the treewidth of the Utility Dependency Graph.

## B   LINEAR PROGRAMMING FOR SOLVING (MASE)

(MASE) can be solved by the following linear programming

$$
\min_{\pi, w} w
$$

$$
w \geq \frac{1}{|S|} \sum_{i \in S} \mathbb{E}_{\boldsymbol{a} \sim \pi} \left[ \mathcal{U}_i \left( \widehat{\boldsymbol{a}}_S, \boldsymbol{a}_{-S} \right) - \mathcal{U}_i \left( \boldsymbol{a} \right) \right] \quad \forall S \in \mathcal{S}, \widehat{\boldsymbol{a}}_S \in \mathcal{A}_S
$$

$$
\pi(\boldsymbol{a}) \geq 0 \quad \forall \boldsymbol{a} \in \mathcal{A}
$$

$$
\sum_{\boldsymbol{a} \in \mathcal{A}} \pi(\boldsymbol{a}) = 1.
$$

Since

$$\mathbb{E}_{\boldsymbol{a}\sim\pi}\left[\mathcal{U}_i\left(\widehat{\boldsymbol{a}}_S, \boldsymbol{a}_{-S}\right)\right] = \sum_{\boldsymbol{a}\in\mathcal{A}} \pi(\boldsymbol{a})\mathcal{U}_i(\widehat{\boldsymbol{a}}_S, \boldsymbol{a}_{-S})$$

$$\mathbb{E}_{\boldsymbol{a}\sim\pi}\left[\mathcal{U}_i\left(\boldsymbol{a}\right)\right] = \sum_{\boldsymbol{a}\in\mathcal{A}} \pi(\boldsymbol{a})\mathcal{U}_i(\boldsymbol{a})$$

are linear in $\pi$, the linear programming above is valid. Note that the linear program contains exponentially many variables ($\pi \in \Delta^{\mathcal{A}}$), so its complexity is necessarily exponential in $N$.

## C    OMITTED PROOFS IN SECTION 4

This section presents the omitted proofs in Section 4.

### C.1    PROOF OF THEOREM 4.1

**Theorem 4.1.** Computing $\epsilon$-MASE is `NP-hard`, even when $\mathcal{S}$ only contains singletons (coalitions of size one) and $1/\epsilon$ is polynomial in the number of players.

*Proof.* We will introduce the allocation problem (`NP-hard`) and show that it can be reduced to computing the correlated strong equilibrium.

**Definition C.1** (Allocation Problem). There are $n$ agents and $m$ goods. An assignment $X\colon [m] \to [n]$ is a mapping from each good to an agent. Agent $i$'s utility is $u_i(X)$ for an assignment $X$. A stochastic allocation $\boldsymbol{p} \in \Delta^{[m]^{[n]}}$ is a distribution over all possible assignments. The egalitarian social welfare (ESW) maximization is defined as

$$\max_{\boldsymbol{p}\in\Delta^{[m]^{[n]}}} \min_{i\in[n]} \sum_{X\in[m]^{[n]}} p(X)u_i(X). \tag{C.1}$$

For any allocation problem, we can create a game with $N = n + m$ players. The action set of player $i \le n$ is $\mathcal{A}_i = \{0,1\}$, while the action set of player $j > n$ is $\mathcal{A}_j = [n]$. For any joint action $\boldsymbol{a} \in \mathcal{A}$, $\mathcal{U}_i(\boldsymbol{a}) = u_i(\boldsymbol{a}_{-[n]})$ for $i \le n$ when $a_1 = a_2 = \cdots = a_n$, otherwise $\mathcal{U}_i(\boldsymbol{a}) = -u_i(\boldsymbol{a}_{-[n]})$. For $j > n, \mathcal{U}_j(\boldsymbol{a}) = 0$. Moreover, let $\mathcal{S} = \{\{1\}, \{2\}, \ldots, \{n\}\}$. We further define

$$a := \sum_{\substack{\boldsymbol{a}\in\mathcal{A}:\\a_2=a_3=\cdots=a_n=0}} \pi(\boldsymbol{a})u_i\left(\boldsymbol{a}_{-[n]}\right)$$

$$b := \sum_{\substack{\boldsymbol{a}\in\mathcal{A}:\\a_2=a_3=\cdots=a_n=1}} \pi(\boldsymbol{a})u_i\left(\boldsymbol{a}_{-[n]}\right)$$

$$c := -\sum_{\substack{\boldsymbol{a}\in\mathcal{A}:\\\exists i,j\in[n]\setminus\{1\},a_i\neq a_j}} \pi(\boldsymbol{a})u_i\left(\boldsymbol{a}_{-[n]}\right).$$

Then, the gap of player 1 is lower bounded by

$$\max_{\widehat{a}_1\in\{0,1\}} \mathbb{E}_{\boldsymbol{a}\sim\pi}\left[\mathcal{U}_i\left(\widehat{a}_1, \boldsymbol{a}_{-1}\right) - \mathcal{U}_i\left(\boldsymbol{a}\right)\right] \ge \max(a,b) + c - (a+b+c) = -\min(a,b).$$

The equation holds when $a_1$ is always equal to $a_2$ when $a_2 = a_3 = \cdots = a_n$. Therefore, the optimal strategy $\pi$ should satisfy $\pi_{[n]}((0,\ldots 0)) = \pi_{[n]}((1,\ldots 1)) = \frac{1}{2}$.

Then, (MASE) is equivalent to

$$\min_{\pi_{[m]}\in\Delta^{[m]^{[n]}}} \max_{i\in[n]} \sum_{\boldsymbol{a}\in\mathcal{A}} -\pi(\boldsymbol{a})u_i(\boldsymbol{a}_{-[n]}) = \min_{\pi_{[m]}\in\Delta^{[m]^{[n]}}} \max_{i\in[n]} \sum_{\boldsymbol{a}_{-[n]}\in[m]^{[n]}} -\pi_{-[n]}(\boldsymbol{a}_{-[n]})u_i(\boldsymbol{a}_{-[n]}),$$

which is equivalent to (C.1). Finally, according to Kawase & Sumita (2020, Corollary 1), it is `NP-hard` to approximate (C.1) up to $1 - \frac{1}{e}$.

The hard instance constructed in Kawase & Sumita (2020) satisfies that $\max_i \max_{X\in[m]^{[n]}} u_i(X) = Poly(n,m)$ and the solution to (C.1) is 1. Therefore, $Poly(N, \frac{1}{\epsilon})$ algorithm does not exist unless P=NP for solving an $\epsilon$-MASE. $\square$

## C.2   PROOF OF THEOREM 4.3

*Proof.* According to Lokshtanov et al. (2011), under SETH, $q$-coloring cannot be solved in $\mathcal{O}((q - \zeta)^{\mathrm{tw}} \cdot \mathrm{Poly}(|I|))$ for arbitrary graph $G$, when a tree decomposition of width $tw$ is given.[7] In the sequel, we construct a game such that computing $\frac{1}{N}$-approximate (MASE) is equivalent to determining the $q$-coloring.

For any $q$-coloring problem on $G = (V, E)$, we will construct a game with $N = |V| + |E|$ players. For each player $i \leq |V|$, the action set $\mathcal{A}_i = \{1, 2, \ldots, q\}$ and the utility function $\mathcal{U}_i \equiv 0$ is a constant function equal to zero. For each player $j > |V|$, the action set is $\{1\}$ and the utility function is $\mathcal{U}_j(\boldsymbol{a}) = \mathbb{1}\left(a_{e_{j-|V|,1}} \neq a_{e_{j-|V|,2}}\right)$, where $(e_{j-|V|,1}, e_{j-|V|,2})$ is the $(j - |V|)^{th}$ edge in $E$ and $\mathbb{1}$ is the indicator function (equals one when the argument is true and otherwise zero). In this game, $\mathcal{S} = \{\{i_1, i_2, j + |V|\} \mid e_j = (i_1, i_2)\}$.

Firstly, for any proper coloring $\boldsymbol{c} \in [q]^{|V|}$, the associated pure strategy is $\pi^{\boldsymbol{c}}$, where $\pi_i^{\boldsymbol{c}}(a_i) = 1$ if and only if $a_i = c_i$ and 0 otherwise. It satisfies (MASE). Because for any $\mathcal{S} \ni S = (i_1, i_2, j + |V|)$, the maximum of $\frac{1}{|S|} \sum_{i \in S} \mathcal{U}_i(\pi)$ is $\frac{1}{3}$, which is attained when the colors of node $i_1, i_2$ are different. Therefore, $\pi^{\boldsymbol{c}}$ obtains the maximum for every coalition $S \in \mathcal{S}$, which implies the satisfaction of (MASE).

Secondly, for any joint strategy $\pi \in \Delta^{\mathcal{A}}$ satisfying (MASE), we have

$$\min_{S \in \mathcal{S}} \frac{\sum_{i \in S} \mathcal{U}_i(\pi)}{|S|} \leq \frac{1}{|\mathcal{S}|} \sum_{S \in \mathcal{S}} \frac{\sum_{i \in S} \mathcal{U}_i(\pi)}{|S|} = \sum_{\boldsymbol{a} \in \mathcal{A}} \pi(\boldsymbol{a}) \frac{1}{|\mathcal{S}|} \sum_{S \in \mathcal{S}} \frac{\sum_{i \in S} \mathcal{U}_i(\boldsymbol{a})}{|S|}$$

$$\leq \frac{1}{3} - \frac{1}{3|E|} \sum_{\boldsymbol{a} \in \mathcal{A}} \pi(\boldsymbol{a}) \mathbb{1}\left(\boldsymbol{a}_{[|V|]} \text{ is not a proper coloring}\right).$$

Because there must exist at least one edge with both of its nodes in the same color for any improper coloring. On the other hand, let $\widehat{\pi} = \pi^{\boldsymbol{c}}$ for some proper coloring $\boldsymbol{c}$. Then, for any $S \in \mathcal{S}$, we have

$$\frac{\sum_{i \in S} \mathcal{U}_i(\widehat{\pi})}{|S|} = \frac{1}{3}.$$

Therefore, the approximation error of (MASE) is at least

$$\frac{1}{3|E|} \sum_{\boldsymbol{a} \in \mathcal{A}} \pi(\boldsymbol{a}) \mathbb{1}\left(\boldsymbol{a}_{[|V|]} \text{ is not a proper coloring}\right)$$

for any joint strategy $\pi$. When we get $\frac{1}{9N^2}$ approximation, since $|E| \leq N^2$, we have

$$\sum_{\boldsymbol{a} \in \mathcal{A}} \pi(\boldsymbol{a}) \mathbb{1}\left(\boldsymbol{a}_{[|V|]} \text{ is not a proper coloring}\right) \leq \frac{1}{3}.$$

Therefore, when sampling $\boldsymbol{a} \sim \pi$, we will get a proper coloring with probability at least $\frac{2}{3}$, which is in complexity class RP. As a result, when P=RP, the time complexity of computing (MASE) is at least $O^*(A^{tw(G)})$, where $G$ is the trust graph and $A$ is the size of the maximal action set. □

## D   PROOF OF THEOREM 5.1

**Theorem 5.1** (Efficient Representation). *For any $\epsilon \geq 0$, at least one of the $\epsilon$-MASE can be represented as a linear combination of $\sum_{S \in \mathcal{S}} |S| \cdot A^{\mathrm{tw}(\mathcal{G})+1}$ pure strategies, where $\mathrm{tw}(\mathcal{G})$ is the treewidth of Utility Dependency Graph.*

---

[7]As summarized in Esmer et al. (2024), the proof of the $q$-coloring complexity implicitly implies that the complexity is lower bounded by $\mathcal{O}(q^{tw} \cdot \mathrm{Poly}(|I|))$, even though a tree decomposition of width $tw$ is given. In other words, aside from computing a tree decomposition, the $q$-coloring itself has an intrinsic computational barrier.

*Proof.* Let $D := \sum_{S \in \mathcal{S}} \sum_{i \in S} |\mathcal{A}_{S \cap \mathcal{N}(i)}|$. For any joint strategy $\pi \in \Delta^{\mathcal{A}}$, consider the vector $\boldsymbol{v}^{\pi} \in \mathbb{R}^D$, where

$$v^{\pi}(S, i, \widehat{\boldsymbol{a}}_{S \cap \mathcal{N}(i)}) = \mathbb{E}_{\boldsymbol{a} \sim \pi} \left[ \mathcal{U}_i(\widehat{\boldsymbol{a}}_{S \cap \mathcal{N}(i)}, \boldsymbol{a}_{-(S \cap \mathcal{N}(i))}) - \mathcal{U}_i(\boldsymbol{a}) \right],$$

for any $S \in \mathcal{S}$ and $\widehat{\boldsymbol{a}}_{S \cap \mathcal{N}(i)} \in \mathcal{A}_{S \cap \mathcal{N}(i)}$. By definition of $\boldsymbol{v}^{\pi}$, it is linear in $\pi$. Therefore, the vertex set of $\{\boldsymbol{v}^{\pi} \mid \pi \in \Delta^{\mathcal{A}}\}$ should correspond to a subset of $\Delta^{\mathcal{A}}$'s vertex set, which is the set of all pure strategies.

By Carathéodory's theorem, for any $\pi^* \in \Delta^{\mathcal{A}}$, $\boldsymbol{v}^{\pi^*}$ can be represented as the linear combination of $D + 1$ vertices, which further implies it can be written as a linear combination of $D + 1$ vectors in $\{\boldsymbol{v}^{\pi} \mid \pi \text{ is a pure strategy}\}$. Then, when $\boldsymbol{v}^{\pi^*} = \sum_{k=1}^{D+1} \lambda^k \boldsymbol{v}^{\pi^k}$ with $\boldsymbol{\lambda} \in \Delta^{D+1}$ and $\pi^1, \pi^2, \dots$ are pure strategies, due to the linearity of $\boldsymbol{v}^{\pi}$, we have

$$\boldsymbol{v}^{\pi^*} = \sum_{k=1}^{D+1} \lambda^k \boldsymbol{v}^{\pi^k} = \boldsymbol{v}^{\sum_{k=1}^{D+1} \lambda^k \pi^k}.$$

Finally,

$$\max_{S \in \mathcal{S}} \max_{\widehat{\boldsymbol{a}}_S \in \mathcal{A}_S} \frac{1}{|S|} \sum_{i \in S} \mathbb{E}_{\boldsymbol{a} \sim \pi} \left[ \mathcal{U}_i(\widehat{\boldsymbol{a}}_S, \boldsymbol{a}_{-S}) - \mathcal{U}_i(\boldsymbol{a}) \right]$$

$$= \max_{S \in \mathcal{S}} \max_{\widehat{\boldsymbol{a}}_S \in \mathcal{A}_S} \frac{1}{|S|} \sum_{i \in S} \mathbb{E}_{\boldsymbol{a} \sim \pi} \left[ \mathcal{U}_i(\widehat{\boldsymbol{a}}_{S \cap \mathcal{N}(i)}, \boldsymbol{a}_{-(S \cap \mathcal{N}(i))}) - \mathcal{U}_i(\boldsymbol{a}) \right]$$

$$= \max_{S \in \mathcal{S}} \max_{\widehat{\boldsymbol{a}}_S \in \mathcal{A}_S} \frac{1}{|S|} \sum_{i \in S} v^{\pi^*}(S, i, \widehat{\boldsymbol{a}}_{S \cap \mathcal{N}(i)})$$

$$= \max_{S \in \mathcal{S}} \max_{\widehat{\boldsymbol{a}}_S \in \mathcal{A}_S} \frac{1}{|S|} \sum_{i \in S} v^{\sum_{k=1}^{D+1} \lambda^k \pi^k}(S, i, \widehat{\boldsymbol{a}}_{S \cap \mathcal{N}(i)}).$$

Hence, once $\pi^*$ satisfies (MASE), there exists a linear combination of $D + 1$ pure strategies also satisfying (MASE). Given $\mathcal{N}(i) \leq \text{tw}(\mathcal{G})$, we have $D \leq \sum_{S \in \mathcal{S}} |S| \cdot A^{\text{tw}(\mathcal{G})}$. □

## E   THE OPTIMALITY OF DYNAMIC PROGRAMMING ON TREE DECOMPOSITION

This section shows that the dynamic programming (*e.g.*, (5.4)) will compute the optimality. Let $u_i(\boldsymbol{a}_{\mathcal{N}(i)})$ be the contribution of player $i$'s utility to the final objective. Recall that $\mathcal{N}_i^S := \mathcal{N}(i) \cap S$. Then, in (5.4),

$$u_i(\boldsymbol{a}_{\mathcal{N}(i)}) = -\sum_{\tau=1}^{t} \sum_{S \in \mathcal{S}} \frac{1}{|S|} \sum_{i \in S} \sum_{\widehat{\boldsymbol{a}}_{\mathcal{N}_i^S} \in \mathcal{A}_{\mathcal{N}_i^S}} \mu^{(\tau)} \left(S, \widehat{\boldsymbol{a}}_{\mathcal{N}_i^S}\right) \left(\mathcal{U}_i \left(\widehat{\boldsymbol{a}}_{\mathcal{N}_i^S}, \boldsymbol{a}_{\mathcal{N}(i) \setminus S}\right) - \mathcal{U}_i(\boldsymbol{a}_{\mathcal{N}(i)})\right)$$

at timestep $t$. We consider the following update rule in this section, which generalizes (5.4) and (G.1) (see the proof of Lemma F.3 and Lemma G.1),

$$h(B, \boldsymbol{a}_B) = \sum_{\substack{i \in [N]: \\ i \text{ assigned to } B}} u_i(\boldsymbol{a}_{\mathcal{N}(i)}) + \sum_{B' \in C(B)} \max_{\substack{\boldsymbol{a}'_{B'} \in \mathcal{A}_{B'}: \\ \boldsymbol{a}_{B \cap B'} = \boldsymbol{a}'_{B \cap B'}}} h(B', \boldsymbol{a}'_{B'}). \tag{E.1}$$

In the following, we will show that (E.1) is optimal.

**Lemma E.1.** For any bag $B \in \mathcal{T}$, let $\text{st}(B) := \{i\}_{i \text{ assigned to } B} \cup \bigcup_{B' \in C(B)} \text{st}(B')$ be the set of players assigned to $B$ and bags in its subtree. Then, for any bag $B \in \mathcal{T}$ and $\boldsymbol{a}_B \in \mathcal{A}_B$, we have

$$h(B, \boldsymbol{a}_B) = \max_{\boldsymbol{a}_{-B} \in \mathcal{A}_{-B}} \sum_{i \in \text{st}(B)} u_i(\boldsymbol{a}_{\mathcal{N}(i)}). \tag{E.2}$$

The proof is postponed to the end of this section. Note that for the root bag $B^r$, since $\mathrm{st}(B^r) = [N]$, Lemma E.1 implies that $\max_{\boldsymbol{a}_{B^r} \in \mathcal{A}_{B^r}} h(B^r, \boldsymbol{a}_{B^r}) = \max_{\boldsymbol{a} \in \mathcal{A}} \sum_{i=1}^{N} u_i(\boldsymbol{a}_{\mathcal{N}(i)})$. Therefore, we find the maximum of $\sum_{i=1}^{N} u_i(\boldsymbol{a}_{\mathcal{N}(i)})$, and the optimal joint action $\boldsymbol{a} \in \mathcal{A}$ can be extracted recursively.

Specifically, let

$$
\boldsymbol{a}_{B^r}^* = \underset{\boldsymbol{a}_{B^r} \in \mathcal{A}_{B^r}}{\mathrm{argmax}} \ h(B^r, \boldsymbol{a}_{B^r})
$$
$$
\forall B \in C(B^r), \quad \boldsymbol{a}_{B \setminus B^r}^* = \underset{\boldsymbol{a}_{B \setminus B^r} \in \mathcal{A}_{B \setminus B^r}}{\mathrm{argmax}} \ d\left(B, (\boldsymbol{a}_{B \setminus B^r}, \boldsymbol{a}_{B \cap B^r}^{(t+1)})\right). \tag{E.3}
$$

We will do this recursively until we find the whole $\boldsymbol{a}^* \in \mathcal{A}$. The tie-breaking rule can be arbitrary, and we use the lexicographic order of joint actions for simplicity. Hence, we prove the optimality of the update rule (E.1). $\square$

**Lemma E.1.** For any bag $B \in \mathcal{T}$, let $\mathrm{st}(B) := \{i\}_{i \text{ assigned to } B} \cup \bigcup_{B' \in C(B)} \mathrm{st}(B')$ be the set of players assigned to $B$ and bags in its subtree. Then, for any bag $B \in \mathcal{T}$ and $\boldsymbol{a}_B \in \mathcal{A}_B$, we have

$$
h(B, \boldsymbol{a}_B) = \max_{\boldsymbol{a}_{-B} \in \mathcal{A}_{-B}} \sum_{i \in \mathrm{st}(B)} u_i(\boldsymbol{a}_{\mathcal{N}(i)}). \tag{E.2}
$$

*Proof.* For leaf bags $B$ ($C(B) = \emptyset$), for any joint action $\boldsymbol{a}_B \in \mathcal{A}_B$, we have

$$
h(B, \boldsymbol{a}_B) = \sum_{\substack{i \in [N]: \\ i \text{ assigned to } B}} u_i(\boldsymbol{a}_{\mathcal{N}(i)}) = \sum_{i \in \mathrm{st}(B)} u_i(\boldsymbol{a}_{\mathcal{N}(i)}) \overset{(i)}{=} \max_{\boldsymbol{a}_{-B} \in \mathcal{A}_{-B}} \sum_{i \in \mathrm{st}(B)} u_i(\boldsymbol{a}_{\mathcal{N}(i)}).
$$

$(i)$ is because $\mathcal{N}(i) \subseteq B$ for any $i$ assigned to $B$ by definition. Additionally, since $B$ is a leaf bag, $\mathrm{st}(B) = \{i \in B : i \text{ assigned to } B\}$.

Then, for any bag $B$ with all of its children $B' \in C(B)$ satisfying (E.2), we have

$$
h(B, \boldsymbol{a}_B) = \sum_{\substack{i \in [N]: \\ i \text{ assigned to } B}} u_i(\boldsymbol{a}_{\mathcal{N}(i)}) + \sum_{B' \in C(B)} \max_{\substack{\boldsymbol{a}_{B'}' \in \mathcal{A}_{B'}: \\ \boldsymbol{a}_{B \cap B'} = \boldsymbol{a}_{B \cap B'}'}} h(B', \boldsymbol{a}_{B'}')
$$

$$
\overset{(i)}{=} \sum_{\substack{i \in [N]: \\ i \text{ assigned to } B}} u_i(\boldsymbol{a}_{\mathcal{N}(i)}) + \sum_{B' \in C(B)} \max_{\substack{\boldsymbol{a}_{B'}' \in \mathcal{A}_{B'}: \\ \boldsymbol{a}_{B \cap B'} = \boldsymbol{a}_{B \cap B'}'}} \max_{\boldsymbol{a}_{-B'}' \in \mathcal{A}_{-B'}} \sum_{i \in \mathrm{st}(B')} u_i(\boldsymbol{a}_{\mathcal{N}(i)}')
$$

$$
= \sum_{\substack{i \in [N]: \\ i \text{ assigned to } B}} u_i(\boldsymbol{a}_{\mathcal{N}(i)}) + \sum_{B' \in C(B)} \max_{\substack{\boldsymbol{a}' \in \mathcal{A}: \\ \boldsymbol{a}_{B \cap B'} = \boldsymbol{a}_{B \cap B'}'}} \sum_{i \in \mathrm{st}(B')} u_i(\boldsymbol{a}_{\mathcal{N}(i)}').
$$

$(i)$ uses the induction hypothesis. By **Property 3 of Tree Decomposition**, for any $B' \in C(B)$ and $i \in \mathrm{st}(B')$, $\mathcal{N}(i) \cap (B \setminus B') = \emptyset$. Because for any $i \in \mathrm{st}(B')$, there must be a bag $B''$ in the subtree of $B'$ such that $\mathcal{N}(i) \subseteq B''$, and **Property 3 of Tree Decomposition** will be violated if $\mathcal{N}(i) \cap B \setminus B' \neq \emptyset$. Then, modifying the constraint $\boldsymbol{a}_{B \cap B'} = \boldsymbol{a}_{B \cap B'}'$ to $\boldsymbol{a}_B = \boldsymbol{a}_B'$ will not change the value of $u_i(\boldsymbol{a}_{\mathcal{N}(i)}')$ for any $i \in \mathrm{st}(B')$. Hence,

$$
h(B, \boldsymbol{a}_B) = \sum_{\substack{i \in [N]: \\ i \text{ assigned to } B}} u_i(\boldsymbol{a}_{\mathcal{N}(i)}) + \sum_{B' \in C(B)} \max_{\substack{\boldsymbol{a}' \in \mathcal{A}: \\ \boldsymbol{a}_B = \boldsymbol{a}_B'}} \sum_{i \in \mathrm{st}(B')} u_i(\boldsymbol{a}_{\mathcal{N}(i)}').
$$

Furthermore, by **Property 3 of Tree Decomposition**, for any $B', B'' \in C(B)$ and $i' \in \mathrm{st}(B'), i'' \in \mathrm{st}(B'')$, we have $\mathcal{N}(i') \cap \mathcal{N}(i'') \subseteq B$. Finally,

$$
h(B, \boldsymbol{a}_B) = \sum_{\substack{i \in [N]: \\ i \text{ assigned to } B}} u_i(\boldsymbol{a}_{\mathcal{N}(i)}) + \max_{\substack{\boldsymbol{a}' \in \mathcal{A}: \\ \boldsymbol{a}_B = \boldsymbol{a}_B'}} \sum_{B' \in C(B)} \sum_{i \in \mathrm{st}(B')} u_i(\boldsymbol{a}_{\mathcal{N}(i)}')
$$

$$
= \max_{\boldsymbol{a}_{-B} \in \mathcal{A}_{-B}} \left( \sum_{\substack{i \in [N]: \\ i \text{ assigned to } B}} u_i(\boldsymbol{a}_{\mathcal{N}(i)}) + \sum_{B' \in C(B)} \sum_{i \in \mathrm{st}(B')} u_i(\boldsymbol{a}_{\mathcal{N}(i)}') \right)
$$

$$
= \max_{\boldsymbol{a}_{-B} \in \mathcal{A}_{-B}} \sum_{i \in \mathrm{st}(B)} u_i(\boldsymbol{a}_{\mathcal{N}(i)}).
$$

This completes the induction. □

## F    PROOF OF THEOREM 5.2

**Theorem 5.2.** Consider (5.3). For any $\delta > 0$, with probability at least $1 - \delta$, the following holds:

$$\max_{\widehat{\pi} \in \Delta^{\mathcal{A}}} \sum_{t=1}^{T} F\left(\pi^{(t)}, \mu^{(t)}\right) - F\left(\widehat{\pi}, \mu^{(t)}\right) \leq 2\left|\mathcal{T}\right| \frac{1 + (\mathrm{tw}(\mathcal{G}) + 1)\log A}{\eta} + 2\eta\left|\mathcal{T}\right| T + \sqrt{2T\log\frac{1}{\delta}}.$$

*Proof.* The proof of Theorem 5.2 can be decomposed into three steps.

Firstly, we show that without loss of generality, if FTPL with a fixed noise $\widetilde{\boldsymbol{n}}$ for all timesteps $t = 1, 2, \ldots$ attains sublinear regret when the adversary is oblivious,[8] then FTPL with independent noise vectors $\widetilde{\boldsymbol{n}}^{(t)}$ also attains the same regret confronting an adaptive adversary. The reduction to the oblivious setting is common in the literature (Agarwal et al., 2019; Suggala & Netrapalli, 2020), and we include it here for completeness.

Secondly, we will show that the regret of a fictitious algorithm $\pi^{(t+1)} \in$ $\mathrm{argmin}_{\pi \in \Delta^{\mathcal{A}}} \sum_{\tau=1}^{(t+1)} F\left(\pi, \mu^{(\tau)}\right) + \left\langle \widetilde{\boldsymbol{n}}^{(t+1)}, \pi \right\rangle$ is sublinear.

Finally, we will show that the regret of (5.3) and that of the fictitious algorithm are close.

### F.1    FIXED NOISE VECTOR

In this section, for completeness, we will show a reduction from an adaptive adversary to an oblivious adversary. For ease of representation, we will take correlator ($\pi$) as the *no-regret learner*, and the deviator ($\mu$) as the *adversary*.

An adaptive adversary determines the utility function at timestep $t$, which is $\mu^{(t)}$ in this section, according to our past strategies, $\pi^{(1)}, \ldots, \pi^{(t-1)}$. In contrast, an oblivious adversary determines all utility functions, *i.e.*, $\mu^{(1)}, \ldots, \mu^{(T)}$, at the beginning (timestep 0), such that $\mu^{(t)}$ is irrelevant to $\pi^{(1)}, \ldots, \pi^{(t-1)}$. In the following, we will show that a sublinear regret against an oblivious adversary implies a sublinear regret against an adaptive adversary.

Intuitively, when the random noise $\widetilde{\boldsymbol{n}}^{(1)}, \ldots, \widetilde{\boldsymbol{n}}^{(T)}$ are independent, $\pi^{(t)}$ only depends on $\mu^{(1)}, \ldots, \mu^{(t-1)}$, which is known to both the oblivious and adaptive adversary, due to the update rule (5.3). Hence, an additional observation on $\pi^{(1)}, \ldots, \pi^{(t-1)}$ does not make adversary more powerful. Formally, we have the following lemma (Cesa-Bianchi & Lugosi, 2006, Lemma 4.1).

**Lemma F.1** (Reformulation of Lemma 4.1 in Cesa-Bianchi & Lugosi (2006)). Consider any randomized no-regret learner and the *distribution* of the decision variable $\pi^{(t)}$ is fully determined by $\mu^{(1)}, \ldots, \mu^{(t-1)}$. Assume the no-regret learner's regret against any sequence of $\mu^{(1)}, \ldots, \mu^{(T)}$ generated by an oblivious adversary satisfies that

$$\underbrace{\mathbb{E}_{\widetilde{\boldsymbol{n}}^{(1)}, \ldots, \widetilde{\boldsymbol{n}}^{(T)}}\left[\max_{\widehat{\pi} \in \Delta^{\mathcal{A}}} \sum_{t=1}^{T} F\left(\pi^{(t)}, \mu^{(t)}\right) - F\left(\widehat{\pi}, \mu^{(t)}\right)\right]}_{\text{Expected Regret}} \leq R.$$

Then, for any sequence of $\mu^{(1)}, \ldots, \mu^{(T)}$ generated by an adaptive adversary and $\delta > 0$, with probability at least $1 - \delta$, we have

$$\max_{\widehat{\pi} \in \Delta^{\mathcal{A}}} \sum_{t=1}^{T} F\left(\pi^{(t)}, \mu^{(t)}\right) - F\left(\widehat{\pi}, \mu^{(t)}\right) \leq R + \sqrt{2T\log\frac{1}{\delta}}.$$

---

[8]An oblivious adversary will choose all the utility functions at timestep 0, while an adaptive adversary will choose the utility functions at timestep $t$ according to $\pi^{(1)}, \pi^{(2)}, \ldots, \pi^{(t-1)}$.

It is easy to see that the distribution of $\pi^{(t)}$ generated by (5.3), whose randomness is induced by $\widetilde{\boldsymbol{n}}^{(t)}$, is fully determined by $\mu^{(1)}, \ldots, \mu^{(t-1)}$, given $\widetilde{\boldsymbol{n}}^{(t)}$ is generated by a fixed distribution independently at each timestep.

Then, we will show that the expected regret of FTPL using independent noise vectors and FTPL with a fixed noise vector is the same, while facing an oblivious adversary.

$$\mathbb{E}_{\widetilde{\boldsymbol{n}}^{(1)}, \ldots, \widetilde{\boldsymbol{n}}^{(T)}}\left[\sum_{t=1}^{T} F\left(\pi^{(t)}, \mu^{(t)}\right)\right] = \sum_{t=1}^{T} \mathbb{E}_{\widetilde{\boldsymbol{n}}^{(1)}, \ldots, \widetilde{\boldsymbol{n}}^{(T)}}\left[F\left(\pi^{(t)}, \mu^{(t)}\right)\right]$$

$$\overset{(i)}{=} \sum_{t=1}^{T} \mathbb{E}_{\widetilde{\boldsymbol{n}}^{(t)}}\left[F\left(\pi^{(t)}, \mu^{(t)}\right)\right].$$

$(i)$ uses the fact that both $\pi^{(t)}$ and $\mu^{(t)}$ are independent of $\widetilde{\boldsymbol{n}}^{(1)}, \ldots, \widetilde{\boldsymbol{n}}^{(t-1)}$, when the adversary controlling $\mu$ is oblivious. Finally, the expectation $\mathbb{E}_{\widetilde{\boldsymbol{n}}^{(1)}}\left[F\left(\pi^{(t)}, \mu^{(t)}\right)\right]$ of

$$\pi^{(t+1)} \in \underset{\pi \in \Delta^{\mathcal{A}}}{\arg\min} \sum_{\tau=1}^{t} F\left(\pi, \mu^{(\tau)}\right) - \left\langle \widetilde{\boldsymbol{n}}^{(1)}, \pi \right\rangle$$

is equal to $\mathbb{E}_{\widetilde{\boldsymbol{n}}^{(t)}}\left[F\left(\pi^{(t)}, \mu^{(t)}\right)\right]$, when $\boldsymbol{n}^{(t)}$ and $\boldsymbol{n}^{(1)}$ are sampled from an identical distribution. In summary,

> fixed noise and an oblivious adversary (Expectation)
> $\Rightarrow$ independent noise and an oblivious adversary (Expectation)
> $\Rightarrow$ independent noise and an adaptive adversary (High Probability Bound).

Hence, the problem reduces to proving sublinear regret against an oblivious adversary with a fixed noise vector for all timesteps.

## F.2 LOW REGRET WITH ACCURATE PREDICTION

The discussion above suggests that we only need to show the sublinear regret against an oblivious adversary, when all timesteps share the same noise vector. In other words, we consider the following update rule,

$$\pi^{(t+1)} \in \underset{\pi \in \Delta^{\mathcal{A}}}{\arg\min} \sum_{\tau=1}^{t} F\left(\pi, \mu^{(\tau)}\right) - \langle \widetilde{\boldsymbol{n}}, \pi \rangle, \tag{F.1}$$

where $\widetilde{\boldsymbol{n}}, \widetilde{\boldsymbol{n}}^{(1)}, \ldots, \widetilde{\boldsymbol{n}}^{(T)}$ are identically distributed.

Next, we will show that if the regret minimizer can make the decision $\pi^{(t+1)}$ with an accurate prediction of $\mu^{(t+1)}$, then we can achieve sublinear regret. In particular, the decision variable at timestep $t + 1$ is chosen according to (F.2).

$$\pi^{(t+1)} \in \underset{\pi \in \Delta^{\mathcal{A}}}{\arg\min} \sum_{\tau=1}^{t+1} F\left(\pi, \mu^{(\tau)}\right) - \langle \widetilde{\boldsymbol{n}}, \pi \rangle. \tag{F.2}$$

Actually, we can see that (F.2) is exactly the original update rule of $\pi^{(t+2)}$. Therefore, we will prove the following lemma in the sequel.

**Lemma F.2.** Consider (F.2). For any timestep $t = 1, 2, \ldots, \mu^{(1)}, \mu^{(2)}, \ldots$, and any $\widehat{\pi} \in \Delta^{\mathcal{A}}$, we have

$$\sum_{\tau=1}^{t}\left(F\left(\pi^{(\tau+1)}, \mu^{(\tau)}\right) - F\left(\widehat{\pi}, \mu^{(\tau)}\right)\right) \le \left\langle \widetilde{\boldsymbol{n}}, \pi^{(2)} - \widehat{\pi} \right\rangle. \tag{F.3}$$

*Proof.* We will prove the lemma by induction. When $t = 1$, we have

$$F\left(\pi^{(2)}, \mu^{(1)}\right) - F\left(\widehat{\pi}, \mu^{(1)}\right)$$

$$= \left(F\left(\pi^{(2)}, \mu^{(1)}\right) - \left\langle \widetilde{\boldsymbol{n}}, \pi^{(2)} \right\rangle\right) - \left(F\left(\widehat{\pi}, \mu^{(1)}\right) - \langle \widetilde{\boldsymbol{n}}, \widehat{\pi} \rangle\right) + \left\langle \widetilde{\boldsymbol{n}}, \pi^{(2)} - \widehat{\pi} \right\rangle$$

$$\overset{(i)}{\le} \left\langle \widetilde{\boldsymbol{n}}, \pi^{(2)} - \widehat{\pi} \right\rangle.$$

$(i)$ is because $\pi^{(2)} \in \arg\min_{\pi \in \Delta^{\mathcal{A}}} F\left(\pi, \mu^{(1)}\right) - \langle \widetilde{\boldsymbol{n}}, \pi \rangle$.

Next, we will show that when (F.3) holds for $t = t_0$, then it also holds for $t = t_0 + 1$. For any $\widehat{\pi} \in \Delta^{\mathcal{A}}$, we have

$$\sum_{\tau=1}^{t_0+1} \left( F\left(\pi^{(\tau+1)}, \mu^{(\tau)}\right) - F\left(\widehat{\pi}, \mu^{(\tau)}\right) \right)$$

$$= \sum_{\tau=1}^{t_0+1} F\left(\pi^{(\tau+1)}, \mu^{(\tau)}\right) - \left( \sum_{\tau=1}^{t_0+1} F\left(\widehat{\pi}, \mu^{(\tau)}\right) - \langle \widetilde{\boldsymbol{n}}, \widehat{\pi} \rangle \right) - \langle \widetilde{\boldsymbol{n}}, \widehat{\pi} \rangle$$

$$\overset{(i)}{\leq} \sum_{\tau=1}^{t_0+1} F\left(\pi^{(\tau+1)}, \mu^{(\tau)}\right) - \left( \sum_{\tau=1}^{t_0+1} F\left(\pi^{(t_0+2)}, \mu^{(\tau)}\right) - \left\langle \widetilde{\boldsymbol{n}}, \pi^{(t_0+2)} \right\rangle \right) - \langle \widetilde{\boldsymbol{n}}, \widehat{\pi} \rangle$$

$$= \sum_{\tau=1}^{t_0} \left( F\left(\pi^{(\tau+1)}, \mu^{(\tau)}\right) - F\left(\pi^{(t_0+2)}, \mu^{(\tau)}\right) \right) + \left\langle \widetilde{\boldsymbol{n}}, \pi^{(t_0+2)} - \widehat{\pi} \right\rangle.$$

$(i)$ is because $\pi^{(t_0+2)} \in \arg\min_{\pi \in \Delta^{\mathcal{A}}} \sum_{\tau=1}^{t_0+1} F\left(\pi, \mu^{(\tau)}\right) - \langle \widetilde{\boldsymbol{n}}, \pi \rangle$. Next, by setting $\widehat{\pi} = \pi^{(t_0+2)}$ in the induction hypothesis, we have

$$\sum_{\tau=1}^{t_0} \left( F\left(\pi^{(\tau+1)}, \mu^{(\tau)}\right) - F\left(\pi^{(t_0+2)}, \mu^{(\tau)}\right) \right) + \left\langle \widetilde{\boldsymbol{n}}, \pi^{(t_0+2)} - \widehat{\pi} \right\rangle$$

$$\leq \left\langle \widetilde{\boldsymbol{n}}, \pi^{(2)} - \pi^{(t_0+2)} \right\rangle + \left\langle \widetilde{\boldsymbol{n}}, \pi^{(t_0+2)} - \widehat{\pi} \right\rangle$$

$$= \left\langle \widetilde{\boldsymbol{n}}, \pi^{(2)} - \widehat{\pi} \right\rangle. \qquad \square$$

### F.3 SUBLINEAR VARIATION

In this section, we will show that the regret of FTPL with/without a prediction of $\mu^{(t+1)}$ is close. Formally, for any $\widehat{\pi} \in \Delta^{\mathcal{A}}$, we have

$$\sum_{t=1}^{T} \left( F\left(\pi^{(t)}, \mu^{(t)}\right) - F\left(\widehat{\pi}, \mu^{(t)}\right) \right)$$

$$= \sum_{t=1}^{T} \left( F\left(\pi^{(t+1)}, \mu^{(t)}\right) - F\left(\widehat{\pi}, \mu^{(t)}\right) \right) + \sum_{t=1}^{T} \left( F\left(\pi^{(t)}, \mu^{(t)}\right) - F\left(\pi^{(t+1)}, \mu^{(t)}\right) \right)$$

$$\overset{(i)}{\leq} \left\langle \widetilde{\boldsymbol{n}}, \pi^{(2)} - \widehat{\pi} \right\rangle + \sum_{t=1}^{T} \left( F\left(\pi^{(t)}, \mu^{(t)}\right) - F\left(\pi^{(t+1)}, \mu^{(t)}\right) \right).$$

$(i)$ uses Lemma F.2.

Moreover, since $\mathcal{U}_i(\boldsymbol{a}) \in [0,1]$ for any $i \in [N]$ and $\boldsymbol{a} \in \mathcal{A}$, we have

$$\max_{\substack{\pi \in \Delta^{\mathcal{A}}, \\ \mu \in \Delta^{\times_{s \in \mathcal{S}} \mathcal{A}_S}}} |F(\pi, \mu)| \leq 1.$$

Then,

$$\mathbb{E}\left[ F\left(\pi^{(t)}, \mu^{(t)}\right) - F\left(\pi^{(t+1)}, \mu^{(t)}\right) \right] \leq \Pr_{\widetilde{\boldsymbol{n}}}\left( \pi^{(t)} \neq \pi^{(t+1)} \right).$$

Hence, we only need to lower bound $\Pr_{\widetilde{\boldsymbol{n}}}\left( \pi^{(t)} = \pi^{(t+1)} \right)$ in the sequel. Recall that $\pi^{(t)}$ is a pure strategy with $\pi^{(t)}\left(\boldsymbol{a}^{(t)}\right) = 1$ for some joint action $\boldsymbol{a}^{(t)} \in \mathcal{A}$. For any bag $B \in \mathcal{T}$, let $fa(B)$ denote its father ($fa(B) = \emptyset$ if $B$ is the root). Then,

$$\Pr_{\widetilde{\boldsymbol{n}}}\left( \pi^{(t)} = \pi^{(t+1)} \right) = \Pr_{\widetilde{\boldsymbol{n}}}\left( \boldsymbol{a}^{(t)} = \boldsymbol{a}^{(t+1)} \right)$$

$$\overset{(i)}{=} \prod_{B \in \mathcal{T}} \Pr_{\widetilde{\boldsymbol{n}}}\left( \boldsymbol{a}^{(t)}_{B \backslash fa(B)} = \boldsymbol{a}^{(t+1)}_{B \backslash fa(B)} \mid \boldsymbol{a}^{(t)}_{fa(B)} = \boldsymbol{a}^{(t+1)}_{fa(B)} \right).$$

According to (5.5), each bag $B \in \mathcal{T}$ only determines $\boldsymbol{a}^{(t)}_{B \setminus fa(B)}$. Since $\boldsymbol{n}^{(t)}(B, \cdot)$ is sampled independently for every bag $B$, it follows that $\boldsymbol{a}^{(t)}_{B' \setminus B}, \boldsymbol{a}^{(t)}_{B'' \setminus B}$ are independent for $B', B'' \in C(B)$ with $B' \neq B''$, by **Property 3 of Tree Decomposition**. Hence, $(i)$ holds.

For any $B \in \mathcal{T}$, to lower bound $\Pr_{\widetilde{\boldsymbol{n}}} \left( \boldsymbol{a}^{(t)}_{B \setminus fa(B)} = \boldsymbol{a}^{(t+1)}_{B \setminus fa(B)} \mid \boldsymbol{a}^{(t)}_{fa(B)} = \boldsymbol{a}^{(t+1)}_{fa(B)} \right)$, we will first get its lower bound while further conditioning on $\boldsymbol{a}^{(t)}_{B \cup fa(B)}$'s value and $\boldsymbol{n}(B, \cdot)$'s value.

Then, $\Pr_{\widetilde{\boldsymbol{n}}} \left( \boldsymbol{a}^{(t)}_{B \setminus fa(B)} = \boldsymbol{a}^{(t+1)}_{B \setminus fa(B)} \mid \boldsymbol{a}^{(t)}_{fa(B)} = \boldsymbol{a}^{(t+1)}_{fa(B)} \right)$ is equal to this conditioned probability integrating over all possible values of $\boldsymbol{a}^{(t)}_{B \cup fa(B)}$ and $\boldsymbol{n}(B, \cdot)$. Formally, we want to lower bound

$$
p^{(t)}_B(\boldsymbol{a}'_{B \cup fa(B)}, \boldsymbol{x}) := \Pr \left( \boldsymbol{a}^{(t+1)}_{B \setminus fa(B)} = \boldsymbol{a}'_{B \setminus fa(B)} \ \middle| \ \boldsymbol{a}^{(t)}_{fa(B)} = \boldsymbol{a}^{(t+1)}_{fa(B)} , \ \boldsymbol{a}^{(t)}_{B \cup fa(B)} = \boldsymbol{a}'_{B \cup fa(B)} , \right.
$$
$$
\text{and } \forall \boldsymbol{a}_{B \setminus fa(B)} \in \mathcal{A}_{B \setminus fa(B)} \setminus \left\{ \boldsymbol{a}'_{B \setminus fa(B)} \right\},
$$
$$
\left. n(B, (\boldsymbol{a}'_{B \cap fa(B)}, \boldsymbol{a}_{B \setminus fa(B)})) = x((\boldsymbol{a}'_{B \cap fa(B)}, \boldsymbol{a}_{B \setminus fa(B)}))) \right)
$$

for any $\boldsymbol{a}'_{B \cup fa(B)} \in \mathcal{A}_{B \cup fa(B)}$ and $\boldsymbol{x} \in \mathbb{R}^{\mathcal{A}_B}$. Then,

$$
\Pr_{\widetilde{\boldsymbol{n}}} \left( \boldsymbol{a}^{(t)}_{B \setminus fa(B)} = \boldsymbol{a}^{(t+1)}_{B \setminus fa(B)} \mid \boldsymbol{a}^{(t)}_{fa(B)} = \boldsymbol{a}^{(t+1)}_{fa(B)} \right) \geq \inf_{\substack{\boldsymbol{a}'_{B \cup fa(B)} \in \mathcal{A}_{B \cup fa(B)}, \\ \boldsymbol{x} \in \mathbb{R}^{\mathcal{A}_B}}} p^{(t)}_B(\boldsymbol{a}'_{B \cup fa(B)}, \boldsymbol{x}),
$$

since $\Pr_{\widetilde{\boldsymbol{n}}} \left( \boldsymbol{a}^{(t)}_{B \setminus fa(B)} = \boldsymbol{a}^{(t+1)}_{B \setminus fa(B)} \mid \boldsymbol{a}^{(t)}_{fa(B)} = \boldsymbol{a}^{(t+1)}_{fa(B)} \right)$ is equal to $p^{(t)}_B(\boldsymbol{a}'_{B \cup fa(B)}, \boldsymbol{x})$ integrating over $\boldsymbol{a}'_{B \cup fa(B)}$ and $\boldsymbol{x}$.

Since $\boldsymbol{a}^{(t)}_{B \setminus fa(B)} = \boldsymbol{a}'_{B \setminus fa(B)}$, for any $\boldsymbol{a}_{B \setminus fa(B)} \in \mathcal{A}_{B \setminus fa(B)} \setminus \left\{ \boldsymbol{a}'_{B \setminus fa(B)} \right\}$, we have

$$
d^{(t)}(B, \boldsymbol{a}'_B) \leq d^{(t)} \left( B, (\boldsymbol{a}'_{B \cap fa(B)}, \boldsymbol{a}_{B \setminus fa(B)}) \right).
$$

This can be equivalently written as

$$
n(B, \boldsymbol{a}'_B) \geq \left( d^{(t)}(B, \boldsymbol{a}'_B) + n(B, \boldsymbol{a}'_B) \right) - d^{(t)} \left( B, (\boldsymbol{a}'_{B \cap fa(B)}, \boldsymbol{a}_{B \setminus fa(B)}) \right).
$$

Then, $\boldsymbol{a}^{(t+1)}_{B \setminus fa(B)} = \boldsymbol{a}'_{B \setminus fa(B)}$ is equivalent to

$$
n(B, \boldsymbol{a}'_B) \geq \left( d^{(t+1)}(B, \boldsymbol{a}'_B) + n(B, \boldsymbol{a}'_B) \right) - d^{(t+1)} \left( B, (\boldsymbol{a}'_{B \cap fa(B)}, \boldsymbol{a}_{B \setminus fa(B)}) \right) \tag{F.4}
$$
$$
= \left( d^{(t)}(B, \boldsymbol{a}'_B) + n(B, \boldsymbol{a}'_B) \right) - d^{(t)} \left( B, (\boldsymbol{a}'_{B \cap fa(B)}, \boldsymbol{a}_{B \setminus fa(B)}) \right)
$$
$$
+ \left( d^{(t)} \left( B, (\boldsymbol{a}'_{B \cap fa(B)}, \boldsymbol{a}_{B \setminus fa(B)}) \right) - d^{(t+1)} \left( B, (\boldsymbol{a}'_{B \cap fa(B)}, \boldsymbol{a}_{B \setminus fa(B)}) \right) \right)
$$
$$
+ \left( d^{(t+1)}(B, \boldsymbol{a}'_B) - d^{(t)}(B, \boldsymbol{a}'_B) \right).
$$

for any $\boldsymbol{a}_{B \setminus fa(B)} \in \mathcal{A}_{B \setminus fa(B)}$. In Lemma F.3, we show that the variation of $\boldsymbol{d}$ is bounded by 1. Therefore,

$$
n(B, \boldsymbol{a}'_B) \geq \left( d^{(t)}(B, \boldsymbol{a}'_B) + n(B, \boldsymbol{a}'_B) \right) - d^{(t)} \left( B, (\boldsymbol{a}'_{B \cap fa(B)}, \boldsymbol{a}_{B \setminus fa(B)}) \right) + 2
$$

implies (F.4).

**Lemma F.3.** Consider the update rule (5.3). For any timestep $t = 1, 2, \ldots, T$, bag $B \in \mathcal{T}$, joint action $\boldsymbol{a}_B \in \mathcal{A}_B$, and noise $\widetilde{\boldsymbol{n}} \in \mathbb{R}^{\times_{B \in \mathcal{T}} \mathcal{A}_B}$, we have

$$
\left| d^{(t+1)}(B, \boldsymbol{a}_B) - d^{(t)}(B, \boldsymbol{a}_B) \right| \leq 1.
$$

The proof is postponed to the end of this section. Let

$$
w = \max_{\boldsymbol{a}_{B\setminus fa(B)} \in \mathcal{A}_{B\setminus fa(B)} \setminus \left\{ \boldsymbol{a}'_{B\setminus fa(B)} \right\}} \left( d^{(t)}(B, \boldsymbol{a}'_B) + n(B, \boldsymbol{a}'_B) \right) - d^{(t)}\left( B, (\boldsymbol{a}'_{B\cap fa(B)}, \boldsymbol{a}_{B\setminus fa(B)}) \right).
$$

Note that $w$ only depends on $\mu^{(1)}, \ldots, \mu^{(t-1)}$ and $\boldsymbol{x}$. Then,

$$
\begin{aligned}
p_B^{(t)}(\boldsymbol{a}'_{B\cup fa(B)}, \boldsymbol{x}) &\geq \Pr\left( n(B, \boldsymbol{a}'_B) \geq w + 2 \,\middle|\, n(B, \boldsymbol{a}'_B) \geq w \right) \\
&= \frac{\Pr\left( n(B, \boldsymbol{a}'_B) \geq w + 2 \right)}{\Pr\left( n(B, \boldsymbol{a}'_B) \geq w \right)} \\
&\overset{(i)}{=} \frac{\exp\left( -\eta(w+2) \right)}{\exp\left( -\eta w \right)} \\
&= \exp\left( -2\eta \right).
\end{aligned}
$$

$(i)$ is because $n(B, \boldsymbol{a}'_B) \sim \mathrm{Exp}(\eta)$. Finally, by union bound,

$$
\begin{aligned}
\Pr_{\widetilde{\boldsymbol{n}}}\left( \pi^{(t)} = \pi^{(t+1)} \right) &\geq 1 - \sum_{B \in \mathcal{T}} \left( 1 - p_B^{(t)}(\boldsymbol{a}'_{B\cup fa(B)}, \boldsymbol{x}) \right) \\
&\geq 1 - \sum_{B \in \mathcal{T}} (1 - \exp(-2\eta)) \\
&\geq 1 - \sum_{B \in \mathcal{T}} 2\eta \\
&= 1 - 2\eta \,|\mathcal{T}|.
\end{aligned}
$$

Therefore,

$$
\mathbb{E}\left[ \sum_{t=1}^{T} F\left( \pi^{(t)}, \mu^{(t)} \right) - F\left( \widehat{\pi}, \mu^{(t)} \right) \right]
$$

$$
\leq \mathbb{E}\left[ \left\langle \widetilde{\boldsymbol{n}}, \pi^{(2)} - \widehat{\pi} \right\rangle \right] + \sum_{t=1}^{T} \mathbb{E}\left[ F\left( \pi^{(t)}, \mu^{(t)} \right) - F\left( \pi^{(t+1)}, \mu^{(t)} \right) \right]
$$

$$
\leq \mathbb{E}\left[ \left\langle \widetilde{\boldsymbol{n}}, \pi^{(2)} - \widehat{\pi} \right\rangle \right] + 2\eta \,|\mathcal{T}|\, T.
$$

Since $\exp(x) \geq 1 + x$ for any $x \in \mathbb{R}$, $(1 - \exp(-2\eta \,|\mathcal{T}|)) \leq 2\eta \,|\mathcal{T}|$. Additionally,

$$
\begin{aligned}
\mathbb{E}\left[ \left\langle \widetilde{\boldsymbol{n}}, \pi^{(2)} - \widehat{\pi} \right\rangle \right] &\overset{(i)}{\leq} \mathbb{E}\left[ \|\widetilde{\boldsymbol{n}}\|_\infty \cdot \left\| \pi^{(2)} - \widehat{\pi} \right\|_1 \right] \leq 2\mathbb{E}\left[ \|\widetilde{\boldsymbol{n}}\|_\infty \right] \\
&\leq 2 \sum_{B \in \mathcal{T}} \max_{\boldsymbol{a}_B \in \mathcal{A}_B} n(B, \boldsymbol{a}_B) \\
&\overset{(i)}{\leq} 2 \sum_{B \in \mathcal{T}} \frac{1 + \log |\mathcal{A}_B|}{\eta}.
\end{aligned}
$$

$(i)$ is by Hölder's Inequality. $(ii)$ is because the expectation of the maximum of $n$ i.i.d. random variable sampled from $\mathrm{Exp}(\eta)$ is upper bounded by $\frac{1 + \log n}{\eta}$ (Agarwal et al., 2019). Furthermore, $\log |\mathcal{A}_B| \leq |B| \cdot \log A \leq (\mathrm{tw}(\mathcal{G}) + 1) \log A$. Hence,

$$
\mathbb{E}\left[ \sum_{t=1}^{T} F\left( \pi^{(t)}, \mu^{(t)} \right) - F\left( \widehat{\pi}, \mu^{(t)} \right) \right] \leq 2\,|\mathcal{T}|\, \frac{1 + (\mathrm{tw}(\mathcal{G}) + 1) \log A}{\eta} + 2\eta \,|\mathcal{T}|\, T.
$$

$\square$

### F.4 PROOF OF AUXILIARY LEMMAS

**Lemma F.3.** Consider the update rule (5.3). For any timestep $t = 1, 2, \ldots, T$, bag $B \in \mathcal{T}$, joint action $\boldsymbol{a}_B \in \mathcal{A}_B$, and noise $\widetilde{\boldsymbol{n}} \in \mathbb{R}^{\times_{B\in\mathcal{T}} \mathcal{A}_B}$, we have

$$
\left| d^{(t+1)}(B, \boldsymbol{a}_B) - d^{(t)}(B, \boldsymbol{a}_B) \right| \leq 1.
$$

*Proof.* Recall Lemma E.1. We can add $|\mathcal{T}|$ players as the noise player, each assigned to a bag in $\mathcal{T}$, with $u_i(\boldsymbol{a}_B) = n(B, \boldsymbol{a}_B)$ so that $\mathcal{N}(i) = B$. Recall that $u_i$ is the contribution of player $i$ to the objective function $F$. Then, by Lemma E.1, for any $B \in \mathcal{T}$ and $\boldsymbol{a}_B \in \mathcal{A}_B$, we have

$$d^{(t)}(B, \boldsymbol{a}_B) = \min_{\boldsymbol{a}_{-B} \in \mathcal{A}_{-B}} \sum_{i \in \mathrm{st}(B)} u_i^{(t)}(\boldsymbol{a}_{\mathcal{N}(i)}),$$

where

$$u_i^{(t)}(\boldsymbol{a}_{\mathcal{N}(i)}) = -\sum_{\tau=1}^{t} \sum_{\substack{S \in \mathcal{S}: \\ i \in S}} \frac{1}{|S|} \sum_{\widehat{\boldsymbol{a}}_{\mathcal{N}_i^S} \in \mathcal{A}_{\mathcal{N}_i^S}} \mu^{(\tau)}\left(S, \widehat{\boldsymbol{a}}_{\mathcal{N}_i^S}\right) \left(\mathcal{U}_i\left(\widehat{\boldsymbol{a}}_{\mathcal{N}_i^S}, \boldsymbol{a}_{\mathcal{N}(i) \setminus S}\right) - \mathcal{U}_i(\boldsymbol{a}_{\mathcal{N}(i)})\right) \text{ for } i \in [N]$$

$$u_i^{(t)}(\boldsymbol{a}_{\mathcal{N}(i)}) = n(B, \boldsymbol{a}_{\mathcal{N}(i)}) \text{ for } i \text{ as the noise player assigned to bag } B.$$

For any noise player, we can see that $u_i^{(t)}(\boldsymbol{a}_{\mathcal{N}(i)}) - u_i^{(t+1)}(\boldsymbol{a}_{\mathcal{N}(i)}) = 0$. For any $i \in [N]$,

$$\left| u_i^{(t)}(\boldsymbol{a}_{\mathcal{N}(i)}) - u_i^{(t+1)}(\boldsymbol{a}_{\mathcal{N}(i)}) \right|$$

$$= \left| \sum_{\substack{S \in \mathcal{S}: \\ i \in S}} \frac{1}{|S|} \sum_{\widehat{\boldsymbol{a}}_{\mathcal{N}_i^S} \in \mathcal{A}_{\mathcal{N}_i^S}} \mu^{(t+1)}\left(S, \widehat{\boldsymbol{a}}_{\mathcal{N}_i^S}\right) \left(\mathcal{U}_i\left(\widehat{\boldsymbol{a}}_{\mathcal{N}_i^S}, \boldsymbol{a}_{\mathcal{N}(i) \setminus S}\right) - \mathcal{U}_i(\boldsymbol{a}_{\mathcal{N}(i)})\right) \right|$$

$$\leq \sum_{\substack{S \in \mathcal{S}: \\ i \in S}} \frac{1}{|S|} \sum_{\widehat{\boldsymbol{a}}_{\mathcal{N}_i^S} \in \mathcal{A}_{\mathcal{N}_i^S}} \mu^{(t+1)}\left(S, \widehat{\boldsymbol{a}}_{\mathcal{N}_i^S}\right) \left| \mathcal{U}_i\left(\widehat{\boldsymbol{a}}_{\mathcal{N}_i^S}, \boldsymbol{a}_{\mathcal{N}(i) \setminus S}\right) - \mathcal{U}_i(\boldsymbol{a}_{\mathcal{N}(i)}) \right|$$

$$\overset{(i)}{\leq} \sum_{\substack{S \in \mathcal{S}: \\ i \in S}} \frac{1}{|S|} \sum_{\widehat{\boldsymbol{a}}_{\mathcal{N}_i^S} \in \mathcal{A}_{\mathcal{N}_i^S}} \mu^{(t+1)}\left(S, \widehat{\boldsymbol{a}}_{\mathcal{N}_i^S}\right).$$

$(i)$ is because $\mathcal{U}_i(\boldsymbol{a}_{\mathcal{N}(i)}) \in [0, 1]$ for any $\boldsymbol{a}_{\mathcal{N}(i)} \in \mathcal{A}_{\mathcal{N}(i)}$.

Recall that by definition, $\boldsymbol{a}_{-B}^{(t)} = \mathrm{argmax}_{\boldsymbol{a}_{-B} \in \mathcal{A}_{-B}} \sum_{i \in \mathrm{st}(B)} u_i^{(t)}(\boldsymbol{a}_{\mathcal{N}(i)})$. Then,

$$d^{(t)}(B, \boldsymbol{a}_B)$$

$$= \sum_{i \in \mathrm{st}(B)} u_i^{(t)}\left(\left(\boldsymbol{a}_{\mathcal{N}(i) \cap B}, \boldsymbol{a}_{\mathcal{N}(i) \setminus B}^{(t)}\right)\right)$$

$$\geq \sum_{i \in \mathrm{st}(B)} u_i^{(t)}\left(\left(\boldsymbol{a}_{\mathcal{N}(i) \cap B}, \boldsymbol{a}_{\mathcal{N}(i) \setminus B}^{(t+1)}\right)\right)$$

$$= \sum_{i \in \mathrm{st}(B)} u_i^{(t+1)}\left(\left(\boldsymbol{a}_{\mathcal{N}(i) \cap B}, \boldsymbol{a}_{\mathcal{N}(i) \setminus B}^{(t+1)}\right)\right) + \left(u_i^{(t+1)}\left(\left(\boldsymbol{a}_{\mathcal{N}(i) \cap B}, \boldsymbol{a}_{\mathcal{N}(i) \setminus B}^{(t+1)}\right)\right) - u_i^{(t)}\left(\left(\boldsymbol{a}_{\mathcal{N}(i) \cap B}, \boldsymbol{a}_{\mathcal{N}(i) \setminus B}^{(t+1)}\right)\right)\right)$$

$$\geq \sum_{i \in \mathrm{st}(B)} \left(u_i^{(t+1)}\left(\left(\boldsymbol{a}_{\mathcal{N}(i) \cap B}, \boldsymbol{a}_{\mathcal{N}(i) \setminus B}^{(t+1)}\right)\right) - \sum_{\substack{S \in \mathcal{S}: \\ i \in S}} \frac{1}{|S|} \sum_{\widehat{\boldsymbol{a}}_{\mathcal{N}_i^S} \in \mathcal{A}_{\mathcal{N}_i^S}} \mu^{(t+1)}\left(S, \widehat{\boldsymbol{a}}_{\mathcal{N}_i^S}\right)\right)$$

$$= d^{(t+1)}(B, \boldsymbol{a}_B) - \sum_{S \in \mathcal{S}} \frac{1}{|S|} \sum_{i \in S \cap \mathrm{st}(B)} \sum_{\widehat{\boldsymbol{a}}_{\mathcal{N}_i^S} \in \mathcal{A}_{\mathcal{N}_i^S}} \mu^{(t+1)}\left(S, \widehat{\boldsymbol{a}}_{\mathcal{N}_i^S}\right)$$

$$\geq d^{(t+1)}(B, \boldsymbol{a}_B) - 1.$$

Similarly, we can get the upper bound that $d^{(t)}(B, \boldsymbol{a}_B) \leq d^{(t+1)}(B, \boldsymbol{a}_B) + 1$. Hence, the proof is completed. □

## G  EFFICIENT UPDATE OF $\mu$

This section provides the omitted details regarding the update procedure for $\mu$ and presents the complete proof of Theorem 5.3.

### G.1 EFFICIENT UPDATE OF $\mu$

The procedure for updating $\mu$ closely parallels that of $\pi$. Specifically, we iterate over all coalitions $S \in \mathcal{S}$ and, for each $S$, determine the optimal action $\boldsymbol{a}_S \in \mathcal{A}_S$. To achieve this, we maintain a dynamic programming vector $\boldsymbol{g}_S \in \mathbb{R}^{\times_{B \in \mathcal{T}} \mathcal{A}_B}$ for each $S \in \mathcal{S}$, which is updated according to

$$
\begin{aligned}
g_S^{(t+1)}(B, \widehat{\boldsymbol{a}}_B) =& \frac{1}{|S|} \sum_{\tau=1}^{t} \sum_{\substack{i \in S: \\ i \text{ assigned to } B}} \sum_{\boldsymbol{a} \in \mathcal{A}} \pi^{(\tau)}(\boldsymbol{a}) \left( \mathcal{U}_i \left( \widehat{\boldsymbol{a}}_{B \cap S}, \boldsymbol{a}_{B \setminus S} \right) - \mathcal{U}_i \left( \boldsymbol{a}_B \right) \right) \\
&+ \sum_{B' \in C(B)} \max_{\substack{\widehat{\boldsymbol{a}}'_{B'} \in \mathcal{A}_{B'}: \\ \widehat{\boldsymbol{a}}_{B \cap B'} = \widehat{\boldsymbol{a}}'_{B \cap B'}}} g^{(t+1)}(B', \widehat{\boldsymbol{a}}'_{B'}) + m^{(t+1)}(B, \widehat{\boldsymbol{a}}_B).
\end{aligned}
$$

At first sight, the $\sum_{\boldsymbol{a} \in \mathcal{A}}$ appears computationally prohibitive, since $\mathcal{A}$ is exponentially large. Fortunately, the update becomes tractable once we recall that $\pi^{(\tau)}$ is always a pure strategy for $\tau \geq 1$. Denote by $\boldsymbol{a}^{(\tau)}$ the joint action selected by $\pi^{(\tau)}$. Then (G.0) simplifies to

$$
\begin{aligned}
g_S^{(t+1)}(B, \widehat{\boldsymbol{a}}_B) =& \frac{1}{|S|} \sum_{\tau=1}^{t} \sum_{\substack{i \in S: \\ i \text{ assigned to } B}} \left( \mathcal{U}_i \left( \widehat{\boldsymbol{a}}_{B \cap S}, \boldsymbol{a}_{B \setminus S}^{(\tau)} \right) - \mathcal{U}_i \left( \boldsymbol{a}_B^{(\tau)} \right) \right) \\
&+ \sum_{B' \in C(B)} \max_{\substack{\widehat{\boldsymbol{a}}'_{B'} \in \mathcal{A}_{B'}: \\ \widehat{\boldsymbol{a}}_{B \cap B'} = \widehat{\boldsymbol{a}}'_{B \cap B'}}} g^{(t+1)}(B', \widehat{\boldsymbol{a}}'_{B'}) + m_S^{(t+1)}(B, \widehat{\boldsymbol{a}}_B).
\end{aligned}
\tag{G.1}
$$

After completing the dynamic programming updates, we focus on the root bag $B^r$ of the tree decomposition. The selected coalition is then $S^{(t+1)} = \operatorname{argmax}_{S \in \mathcal{S}} \max_{\widehat{\boldsymbol{a}}_{B^r} \in \mathcal{A}_{B^r}} g_S^{(t+1)}(B^r, \widehat{\boldsymbol{a}}_{B^r})$. Next, we apply the reconstruction procedure in (5.5) on $\boldsymbol{g}_{S^{(t+1)}}^{(t+1)}$ to extract a joint action $\widehat{\boldsymbol{a}}^{(t+1)} \in \mathcal{A}$. Finally, we update $\mu^{(t+1)}\left(S^{(t+1)}, \widehat{\boldsymbol{a}}^{(t+1)}\right) = 1$.

This procedure ensures that $\mu$ can be updated efficiently while maintaining consistency with the tree decomposition structure. Analogous to the update of $\pi$, the regret of this process can be bounded.

### G.2 PROOF OF THEOREM 5.3

**Theorem 5.3.** Consider the updates in (5.3). For any $\delta > 0$, with probability at least $1 - \delta$, the following holds:

$$
\max_{\widehat{\mu} \in \Delta^{\times_{S \in \mathcal{S}} \mathcal{A}_S}} \sum_{t=1}^{T} F\left(\pi^{(t)}, \widehat{\mu}\right) - F\left(\pi^{(t)}, \mu^{(t)}\right) \leq 2|\mathcal{T}| \frac{1 + (\text{tw}(\mathcal{G}) + 1) \log A}{\eta} + 2\eta |\mathcal{T}| T + \sqrt{2T \log \frac{1}{\delta}}.
$$

*Proof.* The proof of Theorem 5.3 is similar to that of Theorem 5.2. By using a similar argument as the proof of Theorem 5.2, for any $\widehat{\mu} \in \Delta^{\times_{S \in \mathcal{S}} \mathcal{A}_S}$, we have

$$
\begin{aligned}
\sum_{t=1}^{T} & \left( F\left(\pi^{(t)}, \widehat{\mu}\right) - F\left(\pi^{(t)}, \mu^{(t)}\right) \right) \\
&\leq \left\langle \widetilde{\boldsymbol{m}}, \widehat{\mu} - \mu^{(2)} \right\rangle + \sum_{t=1}^{T} \left( F\left(\pi^{(t)}, \mu^{(t+1)}\right) - F\left(\pi^{(t)}, \mu^{(t)}\right) \right).
\end{aligned}
$$

Next, by introducing the counterpart of Lemma F.3 in the following, the rest of the proof follows that of Theorem 5.2.

**Lemma G.1.** Consider the update rule (5.3). For any timestep $t = 1, 2, \ldots, T$, bag $B \in \mathcal{T}$, joint action $\widehat{\boldsymbol{a}}_B \in \mathcal{A}_B$, coalition $S \in \mathcal{S}$, and noise $\widetilde{\boldsymbol{m}} \in \mathbb{R}^{\times_{B \in \mathcal{T}} \mathcal{A}_B}$, we have

$$
\left| g_S^{(t+1)}(B, \widehat{\boldsymbol{a}}_B) - g_S^{(t)}(B, \widehat{\boldsymbol{a}}_B) \right| \leq 1.
$$

The proof is postponed to the end of this section. Let $S^{(t)}, \widehat{a}^{(t)}$ denote the coalition and action the deviator picks at timestep $t$, *i.e.*, $\mu^{(t)}\left(S^{(t)}, \widehat{a}^{(t)}\right) = 1$. Then,

$$
\mathbb{E}\left[F\left(\pi^{(t)}, \mu^{(t+1)}\right) - F\left(\pi^{(t)}, \mu^{(t)}\right)\right]
$$

$$
\leq \Pr_{\widetilde{m}}\left(\mu^{(t+1)} \neq \mu^{(t)}\right)
$$

$$
= \Pr_{\widetilde{m}}\left(S^{(t)} = S^{(t+1)}\right) \prod_{B \in \mathcal{T}} \Pr_{\widetilde{m}}\left(\widehat{a}_{B \backslash fa(B)}^{(t)} = \widehat{a}_{B \backslash fa(B)}^{(t+1)} \,\middle|\, \widehat{a}_{fa(B)}^{(t)} = \widehat{a}_{fa(B)}^{(t+1)}, S^{(t)} = S^{(t+1)}\right)
$$

$$
\overset{(i)}{\leq} 1 - \exp\left(2\eta|\mathcal{T}|\right).
$$

$(i)$ is because choosing $S^{(t)}$ is equivalent to adding a new player in the root bag $B^r$, whose action is to select the coalition. Finally, for any $\delta > 0$, with probability at least $1 - \delta$,

$$
\sum_{t=1}^{T}\left(F\left(\pi^{(t)}, \widehat{\mu}\right) - F\left(\pi^{(t)}, \mu^{(t)}\right)\right)
$$

$$
\leq 2|\mathcal{T}|\frac{1 + (\mathrm{tw}(\mathcal{G}) + 1)\log A}{\eta} + 2\eta|\mathcal{T}|T + \sqrt{2T\log\frac{1}{\delta}}. \qquad \square
$$

### G.3 Proof of Auxiliary Lemmas

**Lemma G.1.** Consider the update rule (5.3). For any timestep $t = 1, 2, \ldots, T$, bag $B \in \mathcal{T}$, joint action $\widehat{a}_B \in \mathcal{A}_B$, coalition $S \in \mathcal{S}$, and noise $\widetilde{m} \in \mathbb{R}^{\times_{B \in \mathcal{T}} \mathcal{A}_B}$, we have

$$
\left|g_S^{(t+1)}(B, \widehat{a}_B) - g_S^{(t)}(B, \widehat{a}_B)\right| \leq 1.
$$

*Proof.* For any $S \in \mathcal{S}$, the upper bound of $\left|g_S^{(t+1)}(B, \widehat{a}_B) - g_S^{(t)}(B, \widehat{a}_B)\right|$ can be obtained similarly to the proof of Lemma F.3 by choosing

$$
u_i^{(t)}(a_{\mathcal{N}(i)}) = \frac{1}{|S|}\sum_{\tau=1}^{t}\sum_{\substack{i \in S: \\ i \text{ assigned to } B}}\sum_{a \in \mathcal{A}}\pi^{(\tau)}(a)\left(\mathcal{U}_i\left(\widehat{a}_{B \cap S}, a_{B \backslash S}\right) - \mathcal{U}_i\left(a_B\right)\right) \text{ for } i \in [N]
$$

$$
u_i^{(t)}(a_{\mathcal{N}(i)}) = m_S(B, a_{\mathcal{N}(i)}) \text{ for } i \text{ as the noise player assigned to bag } B.
$$

Then,

$$
\left|u_i^{(t)}(a_{\mathcal{N}(i)}) - u_i^{(t+1)}(a_{\mathcal{N}(i)})\right| = \left|\frac{1}{|S|}\sum_{\substack{i \in S: \\ i \text{ assigned to } B}}\sum_{a \in \mathcal{A}}\pi^{(t+1)}(a)\left(\mathcal{U}_i\left(\widehat{a}_{B \cap S}, a_{B \backslash S}\right) - \mathcal{U}_i\left(a_B\right)\right)\right|
$$

$$
\overset{(i)}{\leq} \frac{1}{|S|}\sum_{\substack{i \in S: \\ i \text{ assigned to } B}}\sum_{a \in \mathcal{A}}\pi^{(t+1)}(a).
$$

$(i)$ is by the fact that $\mathcal{U}_i\left(\widehat{a}_{B \cap S}, a_{B \backslash S}\right), \mathcal{U}_i\left(a_B\right) \in [0, 1]$. The rest of the proof follows that of Lemma F.3, and thus we complete the proof. $\qquad \square$

## H Additional Experimental Results

In this section, we detail our experimental setup and report additional results for two further games: the Chicken game (Bergstrom & Godfrey-Smith, 1998) and Pigou's network (Pigou, 2017). All experiments are conducted on a 13th Gen Intel(R) Core(TM) i7-13700K @ 3.40 GHz. Error bars for MASE and FTPL indicate $\pm 1\sigma$ over 100 random seeds $(0, 1, \ldots, 99)$. Across all experiments, we set the learning rate to $\eta = 0.01$ and run for $T = 10,000$ timesteps.

## H.1 EXPERIMENTAL DETAILS

For all baselines, we run each algorithm independently for each player, thus the average strategy converges to a CCE (Hazan et al., 2016). For FTRL, Hedge, and OMD, each player $i \in [N]$ is initialized with a uniform distribution $\pi_i^{(1)}$ over all actions. MASE and FTPL are initialized with a pure strategy chosen uniformly at random from all pure strategies.

## H.2 UTILITY FUNCTIONS

This subsection specifies the utility functions for all four games.

**Prisoner's Dilemma.** The utility matrix is shown in Table 2. If both prisoners confess, they receive reduced sentences. If one confesses while the other defects, the confessor is imprisoned and the defector is released immediately. If both defect, both are imprisoned for longer than in the mutual-confession case to penalize dishonesty.

|  | **Confess (C)** | **Defect (D)** |
|---|---|---|
| **Confess (C)** | $(0.6, 0.6)$ | $(0, 1)$ |
| **Defect (D)** | $(1, 0)$ | $(0.2, 0.2)$ |

Table 2: Utility matrix of the Prisoner's Dilemma. Each entry $(a, b)$ denotes the payoffs to the row player $(a)$ and the column player $(b)$.

**Stag Hunt.** The utility matrix is shown in Table 3. A stag yields a higher reward, but it can only be hunted successfully if both players choose Stag; a solo stag attempt yields nothing. A hare provides a smaller payoff but can be secured by a single player.

|  | **Stag (S)** | **Hare (H)** |
|---|---|---|
| **Stag (S)** | $(1, 1)$ | $(0.1, 0.8)$ |
| **Hare (H)** | $(0.8, 0.1)$ | $(0.5, 0.5)$ |

Table 3: Utility matrix of the Stag Hunt. Each entry $(a, b)$ denotes the payoff of the row player $(a)$ and the column player $(b)$.

**Chicken Game.** Two drivers head toward each other and can either swerve or go straight. If one goes straight while the other swerves, the swerving player "loses." If both go straight, they crash.

|  | **Swerve (Sw)** | **Straight (St)** |
|---|---|---|
| **Swerve (Sw)** | $(5/6, 5/6)$ | $(2/3, 1)$ |
| **Straight (St)** | $(1, 2/3)$ | $(0, 0)$ |

Table 4: Utility matrix of the Chicken game. Each entry $(a, b)$ denotes the payoff of the row player $(a)$ and the column player $(b)$.

**Pigou Network.** We use a three-player variant of Pigou's network. Each player chooses a *fast* or *slow* route. The slow route yields a constant utility of $0.25$. The fast route yields utility $1.5 - 0.5 \cdot$ (number of players choosing the fast route), reflecting congestion.

## H.3 ADDITIONAL EXPERIMENTAL RESULTS

Figure 5 reports additional experiments on the Chicken game and Pigou's network. MASE consistently outperforms the baselines in both coalition exploitability and social welfare. In Pigou's network, purely self-interested players overuse the fast route, which in equilibrium becomes slow. By contrast, when players form coalitions and consider average utility within a coalition, they share the routes so that everyone is better off.

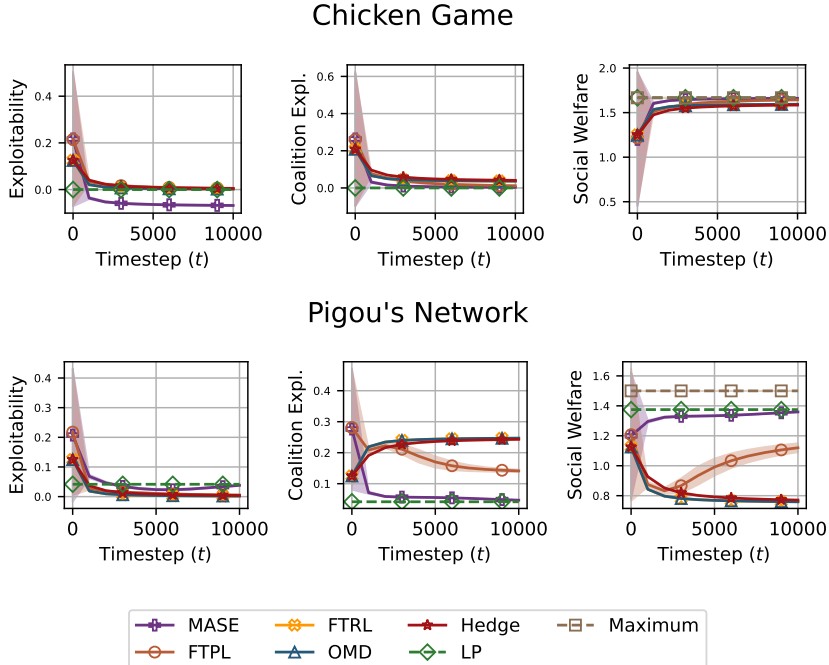

Figure 5: LP refers to the linear program in Section B. Maximum marks the maximum social welfare. MASE outperforms the baselines in both games in terms of coalition exploitability and social welfare.

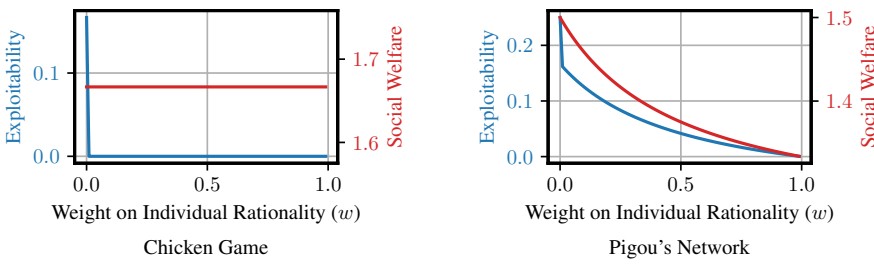

Figure 6: The trade-off between exploitability and social welfare in the Chicken game and the Pigou's network.

Figure 6 shows the trade-off between exploitability and social welfare in the Chicken game and Pigou's network.

Figure 7 reports the runtime of the algorithm for polymatrix games with varying numbers of players, action set sizes, and interaction densities.

## H.4 PROOF OF LEMMA 6.1

**Lemma 6.1.** For any $\epsilon > 0$, computing the CCE with exploitability no more than $\epsilon$ that maximizes social welfare is equivalent to (6.1) by setting $\mathcal{S} = \{\{i\}\}_{i \in [N]} \cup \{[N]\}$ and using the weights:

$$w_S = \begin{cases} w & \text{if } |S| = 1 \\ 1 - w & \text{if } S = [N] \end{cases}$$

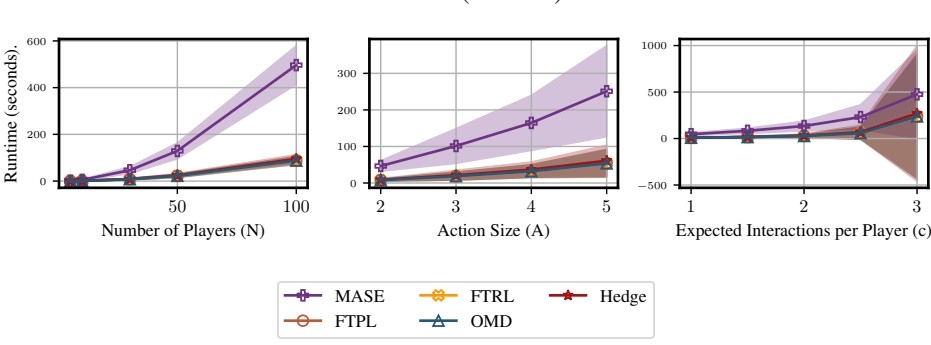

Figure 7: The runtime of the algorithm in different polymatrix games.

for some $w \in [0, 1)$. Conversely, solving (6.1) with these parameters corresponds to finding a point on the Pareto frontier of social welfare and exploitability.

*Proof.* For any $\epsilon > 0$, let $\pi^*$ be the strategy that maximizes social welfare subject to its exploitability being at most $\epsilon$. Let $g^* = \max_{\widehat{a} \in \mathcal{A}} \frac{1}{N} \sum_{i=1}^{N} \mathbb{E}_{\boldsymbol{a} \sim \pi^*} [\mathcal{U}_i(\widehat{\boldsymbol{a}}) - \mathcal{U}_i(\boldsymbol{a})]$ be the maximum gain for the grand coalition $[N]$. Let $w = \frac{g^*}{\epsilon + g^*}$. Then, by construction, the objective value for $\pi^*$ under (6.1) is:

$$\max_{S \in \mathcal{S}} \max_{\widehat{\boldsymbol{a}}_S \in \mathcal{A}_S} \frac{w_S}{|S|} \sum_{i \in S} \mathbb{E}_{\boldsymbol{a} \sim \pi^*}[\ldots] = \max\left(w \cdot (\text{exploitability}), (1-w) \cdot g^*\right)$$

$$\leq \max\left(w\epsilon, (1-w)g^*\right) = \frac{\epsilon g^*}{\epsilon + g^*}.$$

Any strategy $\widehat{\pi}$ with exploitability $> \epsilon$ would have an objective value $> w\epsilon = \frac{\epsilon g^*}{\epsilon + g^*}$, which is worse than the value $\pi^*$ achieves. Therefore, any optimal solution to (6.1) must have exploitability at most $\epsilon$. Since $\pi^*$ by definition maximizes social welfare (i.e., minimizes the coalition gain $g^*$) among all strategies in this set, it must also be an optimal solution to (6.1).

Conversely, for any $w \in [0, 1)$, let $\pi^*$ be the corresponding strategy that optimizes (6.1). Let its exploitability be

$$\epsilon = \max_{i \in [N]} \max_{\widehat{a}_i \in \mathcal{A}_i} \mathbb{E}_{\boldsymbol{a} \sim \pi^*} \left[\mathcal{U}_i(\widehat{a}_i, \boldsymbol{a}_{-i}) - \mathcal{U}_i(\boldsymbol{a})\right].$$

We will show by contradiction that no strategy $\pi'$ exists such that exploitability$(\pi') \leq \epsilon$ and $SW(\pi') > SW(\pi^*)$ (which implies $g' < g^*$, where $g'$ is the gain for the grand coalition under $\pi'$).

Suppose such a $\pi'$ exists. We analyze two cases:

**Case 1: exploitability**$(\pi') < \epsilon$. Since $\pi'$ has both strictly lower exploitability than $\pi^*$ and $g' < g^*$ (higher social welfare), its objective value is $\max(w \cdot \text{exploitability}(\pi'), (1-w)g')$. This is strictly less than $\max(w\epsilon, (1-w)g^*)$, which is the objective value of $\pi^*$. This contradicts the optimality of $\pi^*$.

**Case 2: exploitability**$(\pi') = \epsilon$. If $\epsilon > 0$, choose a small $\delta > 0$ and consider the mixed strategy $\pi_{new} = (1-\delta)\pi' + \delta\pi''$, where $\pi''$ is an arbitrary CCE, which is guaranteed to exist (Nash, 1950). For any $i \in [N]$ and $\widehat{a}_i \in \mathcal{A}_i$, we have:

$$\mathbb{E}_{\boldsymbol{a} \sim \pi_{new}} \left[\mathcal{U}_i(\widehat{a}_i, \boldsymbol{a}_{-i}) - \mathcal{U}_i(\boldsymbol{a})\right]$$
$$= (1-\delta)\mathbb{E}_{\boldsymbol{a} \sim \pi'} \left[\mathcal{U}_i(\widehat{a}_i, \boldsymbol{a}_{-i}) - \mathcal{U}_i(\boldsymbol{a})\right] + \delta\mathbb{E}_{\boldsymbol{a} \sim \pi''} \left[\mathcal{U}_i(\widehat{a}_i, \boldsymbol{a}_{-i}) - \mathcal{U}_i(\boldsymbol{a})\right]$$
$$\overset{(i)}{\leq} (1-\delta)\mathbb{E}_{\boldsymbol{a} \sim \pi'} \left[\mathcal{U}_i(\widehat{a}_i, \boldsymbol{a}_{-i}) - \mathcal{U}_i(\boldsymbol{a})\right]$$
$$\leq (1-\delta)\epsilon.$$

Step $(i)$ holds because $\pi''$ is a CCE, so its exploitability $\mathbb{E}_{\pi''}[\dots]$ is $\leq 0$. Since $\epsilon > 0$, the new strategy $\pi_{new}$ has exploitability$(\pi_{new}) < \epsilon$. By continuity, for sufficiently small $\delta$, $SW(\pi_{new})$ remains strictly higher than $SW(\pi^*)$ (since $SW(\pi') > SW(\pi^*)$). This puts us in Case 1, which leads to a contradiction.

If $\epsilon \leq 0$, then exploitability$(\pi^*) \leq 0$. The objective value for $\pi^*$ is $\max(w\epsilon, (1-w)g^*) = (1-w)g^*$ (since $w\epsilon \leq 0$ and $(1-w)g^* \geq 0$ by definition). The hypothetical strategy $\pi'$ has exploitability$(\pi') = \epsilon \leq 0$ and $g' < g^*$. Its objective value is $\max(w\epsilon, (1-w)g') = (1-w)g'$. Since $g' < g^*$ and $w < 1$, the objective value of $\pi'$ is strictly less than that of $\pi^*$, which contradicts the optimality of $\pi^*$.

In all cases, the existence of such a $\pi'$ leads to a contradiction. Thus, $\pi^*$ must be a solution that maximizes social welfare for a given exploitability $\epsilon$. $\qquad\square$

# I  POLYMATRIX GAMES

In this section, we present experimental details for MASE on games with a larger number of players. We select polymatrix games as the benchmark for these large-scale experiments. This choice is motivated by their inherent graphical structure, which allows for the efficient generation of instances with a low treewidth of their Utility Dependency Graph.

We begin with the formal definition. A polymatrix game has a corresponding undirected graph $\mathcal{G}^U = (\mathcal{V}^U, \mathcal{E}^U)$, with $\mathcal{V}^U = [N]$. For any joint action $\boldsymbol{a} \in \mathcal{A}$, the utility of any player $i$ is defined as:

$$\mathcal{U}_i(\boldsymbol{a}) \coloneqq \sum_{(i,j) \in \mathcal{E}^U} \mathcal{U}_{i,j}(a_i, a_j), \tag{I.1}$$

where $\mathcal{U}_{i,j} \colon \mathcal{A}_i \times \mathcal{A}_j \to [0,1]$ represents the interaction between players $i$ and $j$. In other words, only players who are connected in $\mathcal{G}^U$ interact, and a player's total utility is the summation of these pairwise interactions.

If we construct the Utility Dependency Graph directly, then the tree decomposition may explode unwillingly, *e.g.*, Figure 8 (a). We can see that the treewidth of $\mathcal{G}^U$ is one while the treewidth of the Utility Dependency Graph is three.

Constructing the Utility Dependency Graph directly from the polymatrix game can cause its treewidth to explode. For example, in Figure 8 (a), the original graph $\mathcal{G}^U$ has a treewidth of one, while the resulting Utility Dependency Graph has a treewidth of three.

To prevent this, we construct a strategically equivalent game (note that this new game is *not* a polymatrix game). This construction explicitly models the pairwise interactions as new players:

- For any original edge $(i, j) \in \mathcal{E}^U$, we introduce two edge players, $e_{i,j}$ and $e_{j,i}$.
- Each edge player $e_{i,j}$ has a singleton action set, $\left|\mathcal{A}_{e_{i,j}}\right| = 1$, meaning it has only a single strategy.
- The utility function of an edge player $e_{i,j}$ is defined as the original interaction utility: $\widetilde{\mathcal{U}}_{e_{i,j}} = \mathcal{U}_{i,j}$.
- The utility function of an original vertex player $i$ (one of the original $N$ players) is now a constant zero: $\widetilde{\mathcal{U}}_i \equiv 0$.

This transformation is illustrated in Figure 8 (b). The Utility Dependency Graph for this new game, shown on the right of Figure 8 (b), now has a treewidth of $\max\left(\mathrm{tw}(\mathcal{G}^U), 2\right)$. This method effectively bounds the treewidth and avoids the undesirable explosion.

Next, we show that the new game and the original polymatrix game are strategically equivalent. In other words, for any joint strategy $\pi \in \Delta^{\mathcal{A}}$, the maximum average deviation gain, $\max_{S \in \mathcal{S}} \max_{\widehat{\boldsymbol{a}}_S \in \mathcal{A}_S} \frac{1}{|S|} \sum_{i \in S} \mathbb{E}_{\boldsymbol{a} \sim \pi} [\mathcal{U}_i(\widehat{\boldsymbol{a}}_S, \boldsymbol{a}_{-S}) - \mathcal{U}_i(\boldsymbol{a})]$, does not change. Recall that since the edge players have only a single action, $\pi \in \Delta^{\mathcal{A}}$ (a distribution over the original players' joint actions) is sufficient to specify the joint strategy in both games.

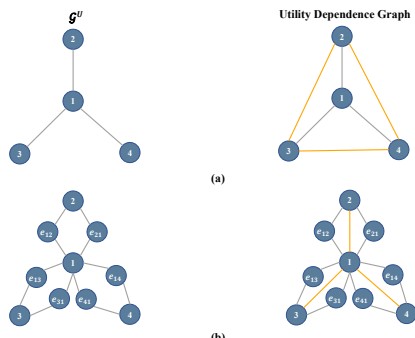

Figure 8: (a) The original graph $\mathcal{G}^U$ corresponding to the polymatrix game (left) and the Utility Dependency Graph. (b) The strategically equivalent game and its Utility Dependency Graph.

**Lemma I.1.** The new game described above is equivalent to the original polymatrix game, when $\widetilde{\mathcal{S}} = \left\{ S \cup \{e_{i,j}\}_{i \in S \wedge (i,j) \in \mathcal{E}^U} \right\}_{S \in \mathcal{S}}$. Formally, for any joint strategy $\pi \in \Delta^{\mathcal{A}}$, we have

$$\max_{S \in \mathcal{S}} \max_{\widehat{\boldsymbol{a}}_S \in \mathcal{A}_S} \frac{1}{|S|} \sum_{i \in S} \mathbb{E}_{\boldsymbol{a} \sim \pi} \left[ \mathcal{U}_i \left( \widehat{\boldsymbol{a}}_S, \boldsymbol{a}_{-S} \right) - \mathcal{U}_i \left( \boldsymbol{a} \right) \right]$$

$$= \max_{\widetilde{S} \in \widetilde{\mathcal{S}}} \max_{\widehat{\boldsymbol{a}}_{\widetilde{S}} \in \mathcal{A}_{\widetilde{S}}} \frac{1}{|\widetilde{S} \cap [N]|} \sum_{i \in \widetilde{S}} \mathbb{E}_{\boldsymbol{a} \sim \pi} \left[ \widetilde{\mathcal{U}}_i \left( \widehat{\boldsymbol{a}}_{\widetilde{S} \cap [N]}, \boldsymbol{a}_{-(\widetilde{S} \cap [N])} \right) - \widetilde{\mathcal{U}}_i \left( \boldsymbol{a} \right) \right].$$

The proof is postponed to the end of this section. This equivalence allows us to solve the new game instead of the original one. Our algorithm can minimize a more general objective, $\max_{S \in \mathcal{S}}$ $\max_{\widehat{\boldsymbol{a}}_S \in \mathcal{A}_S} w_S \sum_{i \in S} \mathbb{E}_{\boldsymbol{a} \sim \pi} \left[ \mathcal{U}_i \left( \widehat{\boldsymbol{a}}_S, \boldsymbol{a}_{-S} \right) - \mathcal{U}_i \left( \boldsymbol{a} \right) \right]$, for any weight vector $\boldsymbol{w} \in \mathbb{R}^S$.[9] We can therefore apply our algorithm to this new, strategically equivalent game.

## I.1 EXPERIMENTAL DETAILS

We generate random polymatrix games using the following procedure:

- Each pair of players $(i, j)$ is connected independently with probability $\frac{c}{N-1}$, where $N$ is the total number of players. This results in an expected degree of $c$ for each player in $\mathcal{G}^U$.

- For each connected pair $(i, j)$, the interaction utilities $\mathcal{U}_{i,j}(a_i, a_j)$ are sampled independently and uniformly from $[0, 1]$ for all action pairs $a_i \in \mathcal{A}_i, a_j \in \mathcal{A}_j$. These pairwise utilities are then normalized according to the formula:

$$\frac{\mathcal{U}_{i,j}(a_i, a_j) - \min_{k \in [N], \widehat{\boldsymbol{a}} \in \mathcal{A}} \mathcal{U}_k(\widehat{\boldsymbol{a}})}{\max_{k \in [N], \widehat{\boldsymbol{a}} \in \mathcal{A}} \mathcal{U}_k(\widehat{\boldsymbol{a}}) - \min_{k \in [N], \widehat{\boldsymbol{a}} \in \mathcal{A}} \mathcal{U}_k(\widehat{\boldsymbol{a}})}.$$

This process ensures that the final total utility $\mathcal{U}_i(\boldsymbol{a})$ for any player $i \in [N]$ and joint action $\boldsymbol{a} \in \mathcal{A}$ falls within the within the range $[0, 1]$.

Consistent with the experiments on small games, we average the results over *100 runs* for each hyper-parameter setting (using seeds 0–99). All algorithms use a learning rate of $\eta = 0.01$, and error bars represent $1 \sigma$. For these larger games, we set the number of timesteps to $T = 100,000$ and a uniform action set size $|\mathcal{A}_i| = A$ for all players $i \in [N]$.

The hyper-parameters for the ablation studies are as follows:

- **Ablation on** $N$: $A = 2$ and $c = 1$.
- **Ablation on** $A$: $N = 30$ and $c = 1$.

---

[9]Both the implementation and the proof only use the linearity of the objective. Hence, any weighted-sum can fit into the framework.

- **Ablation on** $c$: $N = 30$ and $A = 2$.

Furthermore, to accelerate the algorithm, without loss of generality, we only need to consider $\mathcal{S} = \{\{i\}\}_{i \in [N]} \cup \{\{i,j\} \mid i,j \in [N], (i,j) \in \mathcal{G}^U\}$ to minimize the coalition exploitability for any coalitions with no more than two players. In other words, for coalitions of two players, we only need to consider the case when they are connected in $\mathcal{G}^U$. As shown in the following lemma.

**Lemma I.2.** For any joint strategy $\pi \in \Delta^{\mathcal{A}}$, by letting $\mathcal{S} = \{\{i\}\}_{i \in [N]} \cup \{\{i,j\} \mid i,j \in [N] \wedge (i,j) \in \mathcal{G}^U\}$, we have

$$\max_{S \in \mathcal{S}} \max_{\widehat{\boldsymbol{a}}_S \in \mathcal{A}_S} \frac{1}{|S|} \sum_{i \in S} \mathbb{E}_{\boldsymbol{a} \sim \pi} [\mathcal{U}_i (\widehat{\boldsymbol{a}}_S, \boldsymbol{a}_{-S}) - \mathcal{U}_i (\boldsymbol{a})]$$

$$= \max_{\substack{S \in \{\{i\}\}_{i \in [N]} \\ \cup \{\{i,j\} \mid i,j \in [N] \wedge i \neq j\}}} \max_{\widehat{\boldsymbol{a}}_S \in \mathcal{A}_S} \frac{1}{|S|} \sum_{i \in S} \mathbb{E}_{\boldsymbol{a} \sim \pi} [\mathcal{U}_i (\widehat{\boldsymbol{a}}_S, \boldsymbol{a}_{-S}) - \mathcal{U}_i (\boldsymbol{a})].$$

The proof is postponed to the end of this section.

## I.2 PROOF OF THE AUXILIARY LEMMA

**Lemma I.1.** The new game described above is equivalent to the original polymatrix game, when $\widetilde{\mathcal{S}} = \left\{ S \cup \{e_{i,j}\}_{i \in S \wedge (i,j) \in \mathcal{E}^U} \right\}_{S \in \mathcal{S}}$. Formally, for any joint strategy $\pi \in \Delta^{\mathcal{A}}$, we have

$$\max_{S \in \mathcal{S}} \max_{\widehat{\boldsymbol{a}}_S \in \mathcal{A}_S} \frac{1}{|S|} \sum_{i \in S} \mathbb{E}_{\boldsymbol{a} \sim \pi} [\mathcal{U}_i (\widehat{\boldsymbol{a}}_S, \boldsymbol{a}_{-S}) - \mathcal{U}_i (\boldsymbol{a})]$$

$$= \max_{\widetilde{S} \in \widetilde{\mathcal{S}}} \max_{\widehat{\boldsymbol{a}}_{\widetilde{S}} \in \mathcal{A}_{\widetilde{S}}} \frac{1}{|\widetilde{S} \cap [N]|} \sum_{i \in \widetilde{S}} \mathbb{E}_{\boldsymbol{a} \sim \pi} \left[ \widetilde{\mathcal{U}}_i \left( \widehat{\boldsymbol{a}}_{\widetilde{S} \cap [N]}, \boldsymbol{a}_{-(\widetilde{S} \cap [N])} \right) - \widetilde{\mathcal{U}}_i (\boldsymbol{a}) \right].$$

*Proof.* For any $S \in \mathcal{S}$, let $\widetilde{S}$ be its correspondence in $\widetilde{\mathcal{S}}$. Then,

$$\max_{\widehat{\boldsymbol{a}}_{\widetilde{S}} \in \mathcal{A}_{\widetilde{S}}} \frac{1}{|\widetilde{S} \cap [N]|} \sum_{i \in \widetilde{S}} \mathbb{E}_{\boldsymbol{a} \sim \pi} \left[ \widetilde{\mathcal{U}}_i \left( \widehat{\boldsymbol{a}}_{\widetilde{S} \cap [N]}, \boldsymbol{a}_{-(\widetilde{S} \cap [N])} \right) - \widetilde{\mathcal{U}}_i (\boldsymbol{a}) \right]$$

$$\overset{(i)}{=} \max_{\widehat{\boldsymbol{a}}_S \in \mathcal{A}_S} \frac{1}{|S|} \sum_{i \in \widetilde{S}} \mathbb{E}_{\boldsymbol{a} \sim \pi} \left[ \widetilde{\mathcal{U}}_i (\widehat{\boldsymbol{a}}_S, \boldsymbol{a}_{-S}) - \widetilde{\mathcal{U}}_i (\boldsymbol{a}) \right]$$

$$\overset{(ii)}{=} \max_{\widehat{\boldsymbol{a}}_S \in \mathcal{A}_S} \frac{1}{|S|} \sum_{i \in S} \sum_{j : (i,j) \in \mathcal{E}^U} \mathbb{E}_{\boldsymbol{a} \sim \pi} \left[ \widetilde{\mathcal{U}}_{e_{i,j}} (\widehat{\boldsymbol{a}}_S, \boldsymbol{a}_{-S}) - \widetilde{\mathcal{U}}_{e_{i,j}} (\boldsymbol{a}) \right]$$

$$\overset{(iii)}{=} \max_{\widehat{\boldsymbol{a}}_S \in \mathcal{A}_S} \frac{1}{|S|} \sum_{i \in S} \mathbb{E}_{\boldsymbol{a} \sim \pi} [\mathcal{U}_i (\widehat{\boldsymbol{a}}_S, \boldsymbol{a}_{-S}) - \mathcal{U}_i (\boldsymbol{a})].$$

$(i)$ uses the fact that $|\mathcal{A}_{e_{i,j}}| = 1$ and $\widetilde{S} \cap [N] = S$. $(ii)$ is because $\mathcal{U}_i \equiv 0$ for any $i \in [N]$. $(iii)$ is by the definition of $\widetilde{\mathcal{U}}_{e_{i,j}}$ and $\widetilde{S}$. $\square$

**Lemma I.2.** For any joint strategy $\pi \in \Delta^{\mathcal{A}}$, by letting $\mathcal{S} = \{\{i\}\}_{i \in [N]} \cup \{\{i,j\} \mid i,j \in [N] \wedge (i,j) \in \mathcal{G}^U\}$, we have

$$\max_{S \in \mathcal{S}} \max_{\widehat{\boldsymbol{a}}_S \in \mathcal{A}_S} \frac{1}{|S|} \sum_{i \in S} \mathbb{E}_{\boldsymbol{a} \sim \pi} [\mathcal{U}_i (\widehat{\boldsymbol{a}}_S, \boldsymbol{a}_{-S}) - \mathcal{U}_i (\boldsymbol{a})]$$

$$= \max_{\substack{S \in \{\{i\}\}_{i \in [N]} \\ \cup \{\{i,j\} \mid i,j \in [N] \wedge i \neq j\}}} \max_{\widehat{\boldsymbol{a}}_S \in \mathcal{A}_S} \frac{1}{|S|} \sum_{i \in S} \mathbb{E}_{\boldsymbol{a} \sim \pi} [\mathcal{U}_i (\widehat{\boldsymbol{a}}_S, \boldsymbol{a}_{-S}) - \mathcal{U}_i (\boldsymbol{a})].$$

*Proof.* For any disconnected players $i, j$ and $S = \{i, j\}$, we can see that

$$\max_{\widehat{\boldsymbol{a}}_S \in \mathcal{A}_S} \frac{1}{|S|} \sum_{k \in S} \mathbb{E}_{\boldsymbol{a} \sim \pi} \left[ \mathcal{U}_k \left( \widehat{\boldsymbol{a}}_S, \boldsymbol{a}_{-S} \right) - \mathcal{U}_k \left( \boldsymbol{a} \right) \right]$$

$$\leq \max_{\widehat{\boldsymbol{a}}_S \in \mathcal{A}_S} \max_{k \in S} \mathbb{E}_{\boldsymbol{a} \sim \pi} \left[ \mathcal{U}_k \left( \widehat{\boldsymbol{a}}_S, \boldsymbol{a}_{-S} \right) - \mathcal{U}_k \left( \boldsymbol{a} \right) \right]$$

$$= \max_{k \in S} \max_{\widehat{\boldsymbol{a}}_S \in \mathcal{A}_S} \mathbb{E}_{\boldsymbol{a} \sim \pi} \left[ \mathcal{U}_k \left( \widehat{\boldsymbol{a}}_S, \boldsymbol{a}_{-S} \right) - \mathcal{U}_k \left( \boldsymbol{a} \right) \right]$$

$$= \max_{k \in S} \max_{\widehat{\boldsymbol{a}}_S \in \mathcal{A}_S} \mathbb{E}_{\boldsymbol{a} \sim \pi} \left[ \sum_{k' \in [N] : \, (k,k') \in \mathcal{G}^U} \mathcal{U}_{k,k'} \left( \widehat{\boldsymbol{a}}_S, \boldsymbol{a}_{-S} \right) - \mathcal{U}_{k,k'} \left( \boldsymbol{a} \right) \right]$$

$$\overset{(i)}{=} \max_{k \in S} \max_{\widehat{\boldsymbol{a}}_S \in \mathcal{A}_S} \mathbb{E}_{\boldsymbol{a} \sim \pi} \left[ \sum_{k' \in [N] : \, (k,k') \in \mathcal{G}^U} \mathcal{U}_{k,k'} \left( \widehat{\boldsymbol{a}}_k, \boldsymbol{a}_{-k} \right) - \mathcal{U}_{k,k'} \left( \boldsymbol{a} \right) \right]$$

$$= \max_{k \in S} \max_{\widehat{\boldsymbol{a}}_S \in \mathcal{A}_S} \mathbb{E}_{\boldsymbol{a} \sim \pi} \left[ \mathcal{U}_k \left( \widehat{\boldsymbol{a}}_k, \boldsymbol{a}_{-k} \right) - \mathcal{U}_k \left( \boldsymbol{a} \right) \right].$$

$(i)$ is because $k' \notin S$ since $k \in S = \{i, j\}$ and $i, j$ are not connected. Therefore, since the coalition exploitability of $S$ is upper bounded by the maximum of that of coalitions $\{i\}$ and $\{j\}$, we do not need to consider $\{i, j\}$.

Actually, the argument can be generalized to coalitions of any size $M$. If we want to consider the coalition exploitability for coalitions no more than size $M$, then we only need to consider all connected coalitions of size no more than size $M$ by an induction similar to the proof above. □

