# OpenReview forum: "Computing Equilibrium beyond Unilateral Deviation"
_ICLR.cc/2026/Conference — ICLR 2026 Poster_

### Official Review · Reviewer_9CSD · 2025-10-21

**Soundness:** 3
**Presentation:** 3
**Contribution:** 3
**Rating:** 8
**Confidence:** 3

**Summary:**

This paper studies an equilibrium concept describing resistance to multilateral deviations, similar to strong NE and coalition-proof equilibrium. This "Minimal Average-Strong Equilibrium" (MASE) is the strategy $\pi$ which minimizes the utility gain any coalition can gain by any deviation, averaged across the coalition's members. Computing this notion is hard (Theorem 5.1), so a fixed parameter lower bound is provided in terms of strategy space size and the tree width of a tree decomposition of the game's Utility Dependency Graph (Theorem 5.3). The paper then goes on to propose and empirically evaluate an algorithm achieving an $\epsilon$-MASE whose running time matches their lower bound (line 416). The main concepts for this algorithm include a correlator–deviator meta-game with FTPL updates and dynamic programming over a tree decomposition.

**Strengths:**

This paper makes a meaningful contribution to the learning-in-games literature by providing an algorithm that achieves the tight attainable bound for $\epsilon$-MASE, established in Theorem 5.3. The work is original in its framing of MASE as a zero-sum meta-game between correlator and deviator strategies, and in its use of the tree decomposition of the utility dependency graph to express $\sum_{S \in \mathcal{S}}$. This appears to be a novel and potentially generalizable technique. The theoretical analysis is rigorous, and the experimental results effectively support the claims, demonstrating the algorithm's efficacy in computing the LP-based solution. The paper is generally clear and well-organized, though some technical sections could benefit from additional explanation to improve accessibility.

**Weaknesses:**

While the paper's theoretical contributions are strong, several aspects of the exposition could be clarified to improve readability and accessibility. In particular, certain definitions and intuitions are introduced later than they are used, and some algorithmic details are not fully explained. For instance:
- After the definition of MASE (line 165), it would be helpful to explicitly mention that minimizing the deviation gap guarantees the existence of MASE, as this may not be immediately obvious to some readers.
Theorem 5.3 (line 216) invokes tree decomposition before defining it (line 316); moving the definition earlier and expanding the discussion of SETH and BPP = P beyond brief footnotes, perhaps in an appendix, would improve clarity.
- The intuition behind the tree decomposition procedure in Figure 1 (line 334) could be expanded, particularly regarding how property 3 ensures no contradictions arise in Equation (6.5).
- The discussion of local assignments of agents to bags (lines 310–312) could be elaborated.
- The FTPL update process (lines 272, 292, 406) could be clarified—are updates alternating, or is there a more complex schedule? Including a concise description of the full procedure for attaining the convergence bound in Theorem 6.4 would be useful.
- In Section 6.2 (lines 301–307), it would be helpful to explain how Equations (6.4) and (6.5) contribute to minimizing $F(\tilde{\pi}, \mu)$
Overall, these issues primarily concern clarity and presentation rather than the soundness of the work, and addressing them would make the paper significantly easier to follow for a broader audience.


Minor:
- Line 230: doesn't this paper only consider deviations of singleton coalitions? (following line 129 and Thm 5.1)
- Line 237: should size be $|A|^n$?
- Line 292: it might be worth saying up front that Section 6.2 is for $\pi$, while the update for $\mu$ is in Appendix F
- Typo line 421 "Social welfare.:"
- Author comments accidentally revealed on lines 712, 760,
- Accidental red text on lines 900, 971, 987

**Questions:**

The following points reflect areas where additional clarification could help readers better understand the technical contributions and experimental comparisons:

- (Line 145) How does the complexity of LP (Appendix A) compare to your solution (lines 413--417)? On Line 145 when you say, for non-succinct games, MASE can be computed by LP in $O(\star^N)$, this doesn't clarify the case for succinct games.
- Relatedly, it appears in Figure 2 that MASE converges to LP. For my own understanding, does LP correspond with $\pi^\star$ and $\mu^\star$ in Theorem 6.4? Can you please clarify this connection?
- Is the deviation gap (line 189) for CCE equivalent to $E_{a \sim \pi}[U_i(hat~a_{S}, a_{-S}) - U_i(a)]$ in the definition of MASE (line 167)? For the reader's understanding, it may be worth an additional sentence drawing this connection, somewhere on lines 187--194.
- (Line 191) How does your work differ from Anagnostides et al. (2025)? How does Theorem 5.1 differ from their conclusion about minimizing the average gap of CCE across players?
- Relatedly, how does Anagnostides et al.'s solution compare to your experiments in Figure 2?
- Is your technique for identifying the tree decomposition of the utility dependency graph novel? Or is it inspired by other work?

---

> ### Author Response · Authors · 2025-11-26
> **Response to Reviewer 9CSD**
>
> We thank the reviewer for the positive assessment of our theoretical contributions and the rigorous analysis. We appreciate the constructive feedback regarding the exposition. We will incorporate these changes in the final version. Below, we address the specific questions raised.
>
> ## FTPL Clarification
>
> We clarify that the FTPL updates are **simultaneous**, not alternating. The superscript $t$ in our notation denotes the discrete time step where both the correlator and deviator update their strategies based on the history up to $t-1$.
>
> ## Coalition Size
>
> This paper considers coalitions of any size. There is a typo on line 230, and we have corrected it.
>
> ## LP complexity
>
> - As noted in Appendix B, the LP formulation requires a variable for every joint action. Thus, the complexity is $Poly(|\mathcal{A}|) = Poly(A^N)$. This remains exponential in $N$ regardless of whether the game representation is succinct or non-succinct.
> - In contrast, our algorithm is **Fixed-Parameter Tractable (FPT)**. It leverages the sparsity of interactions. When the Utility Dependency Graph has a small treewidth (as is often the case in succinct games or sparse graphs), our algorithm is polynomial in $N$, whereas the LP remains computationally intractable ($O(A^N)$).
>
> ## LP convergence
>
> > "Relatedly, it appears in Figure 2 that MASE converges to LP. For my own understanding, does LP correspond with $\pi^{\star}$ and $\mu^{\star}$ in Theorem 6.4? Can you please clarify this connection?"
>
> Yes, the LP directly solves an MASE objective, so it converges to $\pi^{\star}$. The variables $\mu^{\star}$ in Theorem 6.4 (Theorem 5.4 in the latest version) are auxiliary variables specific to our dynamic programming approach and are not computed by the standard LP.
>
> ## Difference to Anagnostides et al. (2025)
>
> > "(Line 191) How does your work differ from Anagnostides et al. (2025)? How does Theorem 5.1 differ from their conclusion about minimizing the average gap of CCE across players?"
>
> The key difference and our primary contribution is addressing *multilateral* deviations by coalitions of any size (Strong Equilibrium). However, the ACCE discussed in Anagnostides et al. (2025) is still about unilateral deviations (singleton coalitions).
>
> ## Novelty of Tree Decomposition
>
> > "Is your technique for identifying the tree decomposition of the utility dependency graph novel? Or is it inspired by other work?"
>
> The _concept_ of tree decompositions in theoretical computer science is definitely not new. For example, these tools are very common for disaggregating the influence of random variables in graphical models (see also the notion of junction tree, and the junction tree theorem). However, the usage of tree decompositions in the context of equilibria robust to multilateral deviations (and in fact the very notion of MASE) is novel.
>
> ## Minor Issue
>
> - **Action Set Size (Line 237):** $\mathcal{A}$ represents the joint action set, so $|\mathcal{A}| = A^N$.
> - **Red Text (Lines 900, 971):** The red coloring was intended to visually emphasize the $+1$ term in the equations. We will adjust this to standard black formatting in the final version to avoid confusion with editorial comments.

---

### Official Review · Reviewer_SFni · 2025-10-27

**Soundness:** 3
**Presentation:** 2
**Contribution:** 3
**Rating:** 6
**Confidence:** 3

**Summary:**

The paper introduces Minimal Average-Strong Equilibrium (MASE), a solution concept designed to handle coalitional (multilateral) deviations in games. Instead of requiring immunity to every coalition (which often fails to exist), MASE minimizes the maximum average gain any allowed coalition can obtain by deviating from a correlated strategy.

On the theory side, the authors prove NP-hardness even when only single-player deviations are allowed and establish a fixed-parameter lower bound: under SETH, computing (even approximating) MASE must be exponential in the treewidth of a newly defined Utility Dependency Graph.

On the algorithmic side, they show a sparse representation exists for ε-MASE and design a algorithm (up to polynomial factors) that reformulates the problem as a correlator-deviator zero-sum game, solved via no-regret learning (FTPL) combined with dynamic programming over a tree decomposition of the dependency graph. Experiments on classic games (e.g., Prisoner’s Dilemma, Stag Hunt) indicate MASE yields higher robustness to coalition deviations and often better social welfare than standard baselines.

In short, the paper delivers a new coalitional stability notion, a tight complexity characterization tied to treewidth, and a practical algorithm whose complexity matches the theoretical lower bound in its dependence on treewidth.

**Strengths:**

The paper introduces a clear and well-motivated solution concept for coalitional deviations, MASE, which fills the gap left by nonexistence of strong equilibria. It gives tight complexity lower bounds linked to treewidth and then matches them with an algorithm using FTPL plus tree decomposition. The sparse representation result is elegant and enables practical computation. The Utility Dependency Graph provides a clean structural lens that connects theory and algorithms. Experiments show improved coalition robustness and often better social welfare compared to standard baselines.

**Weaknesses:**

About model: using the average gain of coalition members as the metric can mask heterogeneity within the coalition. Large gains for some members can offset small losses for others, which may understate the true threat of coordinated deviations. If one instead imposed per-member constraints or a weighted minimization, both the conclusions and the algorithmic complexity could change. The paper lacks a systematic comparison and robustness analysis along these dimensions.
About result: Although the results regarding complexity in the paper are mathematically solid, they are not surprising.
About writing: in this paper, the authors mention that one can construct a Utility Dependency Graph from the game G. What implicit relationship exists between the two, and what aspects of that relationship should be clarified in the main text?

**Questions:**

First, what is the relationship between your Utility Dependency Graph and the original game? Note that I am not asking how to construct the graph from the game; rather, I want to understand the underlying relationship between the constructed graph and the original game.
Second, why do you rely on a tree decomposition of the Utility Dependency Graph as the core algorithmic vehicle? How do bags correspond to interaction structures or utility terms in the original game?
In a word, I find the description of the tree decomposition and the dynamic programming on the tree insufficiently intuitive, and I would like the authors to clarify it.

---

> ### Author Response · Authors · 2025-11-26
> **Response to Reviewer SFni**
>
> We thank the reviewer for the insightful comments and for recognizing the soundness of our theoretical results and the elegance of the sparse representation. We address the concerns regarding the objective metric, novelty, and the intuition behind the graphical structures below.
>
> ## Average over Coalitions
>
> > "Large gains for some members can offset small losses for others, which may understate the true threat of coordinated deviations. If one instead imposed per-member constraints or a weighted minimization, ..."
>
> We have clarified in the revision that our algorithm and positive results hold for any weighted average objective (not just the uniform average). The core requirement for our approach is the linearity of the objective function, which allows us to cast the problem as a zero-sum meta-game. The "average" used in the initial submission can be viewed as a specific instance of a weighted sum where weights are uniform ($1/|S|$). We have updated the paper to explicitly state that arbitrary weights can be handled without changing the complexity. (Please see also the response to Reviewer D58R.)
>
> ## Novelty of results
>
> > "Although the results regarding complexity in the paper are mathematically solid, they are not surprising."
>
> We respectfully disagree that the results are not surprising. Progress related to strong equilibria in general has resisted progress for a long time. The need for solution concepts robust beyond unilateral deviations is pretty self-evident. Aumann proposed Strong Nash Equilibrium (SNE) in 1959 [1]. However, positive computational results have been virtually non-existent, due to hardness and non-existence issues. Coalition-Proof Equilibrium [2] also attempted to provide a notion suitable for multilateral deviations, but suffers from similar issue. We believe that the solution concept in our paper still makes plenty of sense from an economic point of view (robustness against profit-sharing deviations), but, for the first time, it comes with positive parameter-tractability results.
>
> ## Utility Dependence Graph
>
> The utility dependence graph characterizes the density of interactions among players. An edge in the UDG indicates that the joint actions of the connected players influence at least one player’s utility. Therefore, any players connected in the graph should be analyzed jointly.
>
> ## Tree Decomposition
>
> We rely on tree decomposition because it transforms the cyclic, complex dependencies of the UDG into a hierarchical tree structure, enabling dynamic programming (DP).
>
> - **Meaning of a Bag:** A "bag" in the decomposition represents a **local interaction cluster**. It contains the minimal set of players required to compute a specific component of the total deviation gain.
> - **The DP Logic:** The "separator" property of the tree decomposition guarantees that, given the actions of players in a specific bag (the root of a subtree), the optimization problem within that subtree is conditionally independent of the rest of the graph. This allows us to cache optimal solutions for the subtree and pass only the necessary information up the tree, reducing the complexity from exponential in $N$ (total players) to exponential in treewidth.
>
> [1] Aumann, Robert J. "Acceptable points in general cooperative n-person games." Contributions to the Theory of Games (AM-40) 4 (2016): 287.
>
> [2] Bernheim, B. Douglas, Bezalel Peleg, and Michael D. Whinston. "Coalition-proof nash equilibria i. concepts." Journal of economic theory 42.1 (1987): 1-12.

---

### Official Review · Reviewer_RC8s · 2025-10-27

**Soundness:** 2
**Presentation:** 2
**Contribution:** 2
**Rating:** 4
**Confidence:** 4

**Summary:**

The paper introduces Minimal Average Strong Equilibrium (MASE), a concept extending classical equilibria to account for coalition deviations by minimizing each coalition’s average incentive to deviate. Computing even an approximate MASE (ϵ-MASE) is NP-hard.

Using a Utility Dependency Graph to capture players’ interdependencies, the authors establish a fixed-parameter lower bound: under SETH, computing MASE requires time exponential in the graph’s treewidth.

They propose a Follow-the-Perturbed-Leader (FTPL) learning algorithm via a correlator–deviator meta-game, achieving $O(\sqrt{T})$ regret for both players, with runtime matching the treewidth lower bound.

Experiments are conducted on bimatrix games.

**Strengths:**

1. A novel solution concept, Minimal Average-Strong Equilibrium (MASE), is proposed to addresses coalition deviations.
2. Clear theoretical results of computation complexity.
3. A no-regret learning algorithm is proposed.

**Weaknesses:**

1. The paper appears to overstate its contributions. In the abstract, the authors claim to propose a tractable equilibrium; however, the theoretical results indicate that computing this equilibrium is NP-hard.
2. Although the theoretical results and learning algorithm are designed for multi-player games, the experiments are limited to simple $2\times 2$ bimatrix games.

**Questions:**

How dose the learning algorithm perform in more general games?

---

> ### Author Response · Authors · 2025-11-26
> **Response to Reviewer RC8s**
>
> We thank the reviewer for the constructive feedback and for recognizing the novelty of the MASE concept and the clarity of our theoretical results. Below, we address the concerns regarding the tractability claims and the experimental scope.
>
> ## Complexity of computing MASE
>
> > "The paper appears to overstate its contributions. In the abstract, the authors claim to propose a tractable equilibrium; however, the theoretical results indicate that computing this equilibrium is NP-hard."
>
> We apologize if the abstract created ambiguity regarding "tractability." To clarify: while computing MASE is indeed NP-hard in the general case (as noted in the review), our contribution is establishing that the problem is **Fixed-Parameter Tractable (FPT)** with respect to the treewidth of the interaction graph.
>
> As detailed in **Theorem 4.3**, we establish a fixed-parameter lower bound, showing that the complexity must be exponential in the treewidth (assuming SETH). This aligns with similar hardness results in graph theory, such as those for graph coloring [1]. However, **Theorem 5.4** and the discussion below demonstrate that our algorithm’s runtime matches this lower bound.
>
> We have revised the abstract to avoid confusion.
>
> ## Experiments
>
> > "Although the theoretical results and learning algorithm are designed for multi-player games, the experiments are limited to simple $2\times 2$ matrix games."
>
> > "How does the learning algorithm perform in more general games?"
>
> We appreciate this suggestion. To demonstrate the efficacy of our learning algorithm in general multi-player settings, we have expanded the experimental section in the revised paper.
>
> We added simulations on **polymatrix games with up to 100 players**. These experiments demonstrate that:
>
> 1. The algorithm scales effectively to large-player settings, consistent with our theoretical bounds (Figure 7).
> 2. Our proposed method is significantly more robust to coalition deviations compared to classical no-regret learning algorithms, successfully converging to MASE even in complex interaction structures (Figure 4).
>
> [1] Lokshtanov, Daniel, Dániel Marx, and Saket Saurabh. "Known algorithms on graphs of bounded treewidth are probably optimal." Proceedings of the twenty-second annual ACM-SIAM symposium on Discrete Algorithms. Society for Industrial and Applied Mathematics, 2011.

---

### Official Review · Reviewer_D58R · 2025-11-02

**Soundness:** 3
**Presentation:** 2
**Contribution:** 3
**Rating:** 4
**Confidence:** 4

**Summary:**

The paper proposes Minimal Average-Strong Equilibrium (MASE), an always well-defined correlated strategy that minimizes the average incentive to deviate among all coalitions. The idea is to find a correlated distribution over joint actions that minimizes the coalitions' average gain from deviating (differently from CE and CCE, which only prevent unilateral deviations).
The paper proves that computing an approximate MASE is NP-hard even when 1-player coalitions only. It then proposes treewidth-based lower bound via the Utility Dependency Graph (UDG), showing an exponential dependancy on treewidth under the Strong Exponential Time Hypothesis (SETH).
For the approximation result, the reduction assumes BPP = P to derandomize sampling. The method computes MASE by recasting it as a two-player zero-sum meta-game between a correlator and a deviator, then running Follow-The-Perturbed-Leader (FTPL) for both sides while implementing each best-response step with dynamic programming (DP) over a tree decomposition of the UDG. The running time scales polynomially in the horizon and number of coalitions, and exponentially in one plus the treewidth of the UDG (as expected, which matches the lower bound up to constants).
Experiments on standard games such as Prisoner’s Dilemma, Stag Hunt, Chicken, and a Pigou network show lower coalition exploitability and higher social welfare than standard online-learning baselines and a linear-programming oracle designed to compute a MASE for small-sized games, while keeping unilateral exploitability comparable.

**Strengths:**

The idea of MASE is quite nice and it avoids non-existence issues of strong or coalition-proof equilibria. The link to the treewidth decomposition of the UDG is also rather nice. The analysis is sound and the negative dependancy on the UDG's treewidth is quite interesting. The writing is good and the paper's more technical parts are quite understandable. The whole contribution should be relevant to the game-thereotical/multi-agent literature.

**Weaknesses:**

Experiments are rather limited and consider rather small games. Only matrix and toy congestion games are shown. No larger and more general families such as polymatrix or congestion game instances are considered. Also, no scaling versus the treewidth of the UDG is shown, which is a pity.
Re: the choice of using an average gain: this is reasonable but also possibly arbitrary and not too strongly motivated in my view.
Some reductions rely on the assumption that BPP = P which is rather strong. Its practical implications could be better discussed.
The appendices need a little polish -- they contain some draft notes.

**Questions:**

Why average across coalition members? Can you comnpare average versus minimum or weighted objectives on at least a small examples?

What happens if the coalitions are restricted by size? Can the algorithm and bounds be parameterized by a size bound?

Did you compute the tree decomposition or assume it to be already there? I'd like to see how compute-intensive it is.

Please add polymatrix and larger congestion games, and show the runtime as the number of players and, more importantly, treewidth grows.

---

> ### Author Response · Authors · 2025-11-26
> **Response to Reviewer D58R**
>
> We appreciate the constructive suggestions regarding experiments and the objective function, which we address below.
>
> ## Average over Coalition
>
> > "Why average across coalition members? Can you compare average versus minimum or weighted objectives on at least a small examples?"
>
> In this paper, we consider the average utility of coalition members. This captures deviation scenarios in which coalitions of players might decide that it is in their interest to deviate jointly and **share** (uniformly) the profits of such a deviation.
>
> In fact, our algorithm and positive results hold for _any_ weighted average objective (not just the uniform average). The core requirement for our approach is the linearity of the objective function, which allows us to cast the problem as a zero-sum meta-game. The "average" used in the initial submission can be viewed as a specific instance of a weighted sum where weights are uniform ($1/|S|$). We have updated the paper to explicitly state that arbitrary weights can be handled without changing the complexity.
>
> The case of the minimum (as opposed to average) is quite different, and likely to be intractable, even under fixed treewidth. It was an open question well before our paper, and our paper does not make a dent into it. From an optimization standpoint, for the case of the minimum benefit among coalition members, the corresponding problem is of the form
>
> $\min_{\pi} \max_{S, \hat{a} _ S} \min_{i \in S} [ U_i(\hat{a} _ S, a_{-S}) - U _ i(a) ]$
>
> This results in a min-max-min structure, unlike the weighted sum, and it cannot be solved via the current zero-sum meta-game approach.
>
> One of the major contributions of our paper is to show that if one moves away from the model of the minimum deviations, and switches to that of the _average_ deviation (which has its own natural economic interpretation discussed above), then finding equilibria robust against multilateral deviations becomes (fixed-parameter) tractable.
>
> ## BPP=P
>
> > Some reductions rely on the assumption that BPP = P which is rather strong.
>
> We appreciate the opportunity to clarify the role of this assumption. We differentiate between two cases:
>
> 1. **Exact MASE:** Our lower bounds for computing an *exact* MASE do **not** rely on BPP=P.
> 2. **Approximate MASE:** The assumption is only required for the hardness reduction of *approximate* MASE. Specifically, we reduce to finding a proper coloring. The reduction implies that a proper coloring is a joint action that has a large probability mass in an approximate MASE. Without assuming a structure on how the equilibrium is stored, one must sample joint actions to verify the coloring. To derive a deterministic hardness result (NP-hardness) from this randomized sampling process, we require the derandomization assumption BPP=P.
>
> However, if we assume the equilibrium is stored in a structured format (*e.g.*, a linear combination of pure strategies), the checking process becomes deterministic, and the BPP=P assumption is not needed. We also note that BPP=P is a standard conjecture in complexity theory [1][2] (widely believed to hold due to the existence of pseudorandom generators), and is commonly used to bridge the gap between randomized and deterministic complexity classes. It is as widely believed as P$\neq$NP.
>
> ## Restricted coalition size
>
> > What happens if the coalitions are restricted by size? Can the algorithm and bounds be parameterized by a size bound?
>
> Our hardness results are robust even against severe restrictions on coalition size. As proven in **Theorem 4.1**, computing an approximate MASE is computationally intractable even if we restrict attention to **1-player coalitions** (*i.e.*, standard Nash-like deviations). Since the problem is hard for coalitions of size 1, it remains hard for any larger bound $> 1$. Therefore, parameterizing by coalition size does not yield a polynomial-time algorithm without additional assumptions on the utility structure.
>
> ## Tree Decomposition
>
> Throughout the theoretical analysis (both negative and positive results), we assume that the tree decomposition is provided as part of the input. Computing (approximate) tree decompositions is a well studied problem in its own right. We do not need to open that box. As long as the interaction has a fixed-treewidth structure, our algorithm will work in polynomial time.
>
> ## Larger Experiments
>
> Per your suggestion, we have significantly expanded the experimental section:
>
> - **Polymatrix Games:** We have added a new dataset of random polymatrix games.
> - **Scalability:** We now include results plotting runtime against random graphs with different numbers of players in Figure 7 of the revision.

---

> > ### Author Response · Authors · 2025-11-26
> >
> > [1] Nisan N, Wigderson A. Hardness vs randomness[J]. Journal of computer and System Sciences, 1994, 49(2): 149-167.
> >
> > [2] Impagliazzo R, Wigderson A. P= BPP if E requires exponential circuits: Derandomizing the XOR lemma[C]//Proceedings of the twenty-ninth annual ACM symposium on Theory of computing. 1997: 220-229.

---

### Meta-Review · Area_Chair_rB4b · 2026-01-14

**Summary:**

**Reviewer D58R:** Experiments limited to small games. Use of the average gain rather than minimum or weighted gain is not well motivated. Unclear practical implications. What if the coalition size is limited? How computation-intensive the tree decomposition is.

**Reviewer RC8s:** Tractability claim versus the NP-hardness in computing the equilibrium. Limited experiments to simple $2\times2$ games.

**Reviewer SFni:** Use of the average gain is not well discussed. Relation between the game and its utility dependency graph (UDG) is not clearly described. Reliance on the tree decomposition is not clearly described.

**Reviewer 9CSD:** Room for improvement in readability/accessibility.

**Additional points:**
- I think that it would be helpful to include the definition of the treewidth.
- Line 320: $\mathcal{T}:=B^1,B^2,\ldots,B^K$ → $\mathcal{T}:=\\\{B^1,B^2,\ldots,B^K\\\}$

**Reviewer Concerns:**

- **Limitation of small games in the experiments (D58R, RC8s):** The revised version contains results on larger games in Section 6.2 and Appendix I.
- **Use of average  gain (D58R, RC8s, SFni):** The revised version states that the positive results hold for any weighted average objective.
- **Relation between game and UDG, reliance on tree decomposition (SFni):** I think that the author rebuttal has adequately addressed these concerns.

**Reviewer Scores:**

The initial evaluations of Reviewers SFni, 9CS0 were already on the positive side. I think that main concerns of Reviewers D58R, RC8s have been adequetely addressed in the rebuttal/revision.

---

### Decision · Program_Chairs · 2026-01-26

Accept (Poster)